# Transformers need glasses! 👓
# Information over-squashing in language tasks

**Federico Barbero***
University of Oxford
federico.barbero@cs.ox.ac.uk

**Andrea Banino**
Google DeepMind
abanino@google.com

**Steven Kapturowski**
Google DeepMind
skapturowski@google.com

**Dharshan Kumaran**
Google DeepMind
dkumaran@google.com

**João G.M. Araújo**
Google DeepMind
joaogui@google.com

**Alex Vitvitskyi**
Google DeepMind
avlife@google.com

**Razvan Pascanu**
Google DeepMind
razp@google.com

**Petar Veličković**
Google DeepMind
petarv@google.com

## Abstract

We study how information propagates in decoder-only Transformers, which are the architectural backbone of most existing frontier large language models (LLMs). We rely on a theoretical signal propagation analysis—specifically, we analyse the representations of the last token in the final layer of the Transformer, as this is the representation used for next-token prediction. Our analysis reveals a *representational collapse* phenomenon: we prove that certain distinct sequences of inputs to the Transformer can yield arbitrarily close representations in the final token. This effect is exacerbated by the low-precision floating-point formats frequently used in modern LLMs. As a result, the model is provably unable to respond to these sequences in different ways—leading to errors in, e.g., tasks involving counting or copying. Further, we show that decoder-only Transformer language models can lose sensitivity to specific tokens in the input, which relates to the well-known phenomenon of *over-squashing* in graph neural networks. We provide empirical evidence supporting our claims on contemporary LLMs. Our theory also points to simple solutions towards ameliorating these issues.

## 1 Introduction

In recent years the field of Natural Language Processing (NLP) has been revolutionised through the introduction of Transformer-based architectures [30]. Large Transformers trained on some version of next-token prediction, known as *Large* Language Models (LLMs), have demonstrated impressive performance across different tasks, including conversational agents [10, 19], understanding multi-modal inputs [1], and code completion [16]. Most contemporary LLMs specifically focus on the decoder part of the original Transformer architecture, and are commonly referred to as *decoder-only* Transformers. Consequently, we focus primarily on such models in this paper.

However, despite the impressive performance of Transformers, recent works have uncovered surprising failures that may point to fundamental issues in their architecture. For instance,

---

*Work performed while at Google DeepMind.

38th Conference on Neural Information Processing Systems (NeurIPS 2024).

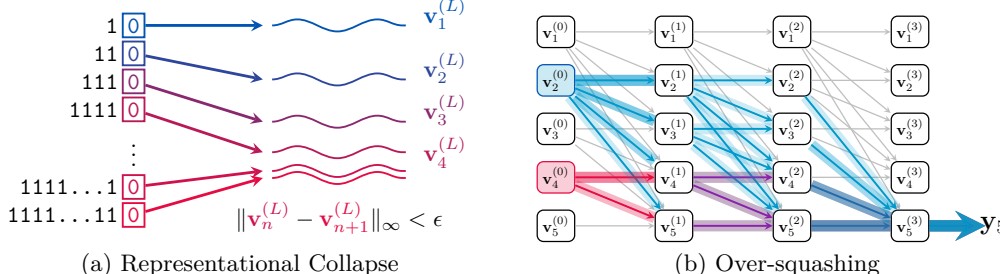

(a) Representational Collapse        (b) Over-squashing

Figure 1: **(a) Representational Collapse** (Theorem 4.2). From top to bottom, we have a series of sequences given to Transformer architectures, each comprising repeated `1` tokens with a single `0` token at the end. The color and proximity of the curved lines illustrate how these representations converge as sequence length increases. **(b) Over-squashing** (Theorem 5.1). Due to the architecture of decoder-only Transformers, tokens that are earlier in their input sequence will have significantly more paths through which their data can reach the representation used for next-token prediction, leading to 'over-squashing'. This effect is depicted here for an early token (blue) and later token (red) in a five-token sequence.

Transformer-based LLMs seem to be particularly challenged by seemingly simple tasks requiring counting [32] or copying elements along input sequences [17]. We find it important to study such failure cases, as these operations are fundamental building blocks of computation, and they are often necessary for solving reasoning tasks. A common strategy to assist LLMs in solving such tasks is to supply them with 'tools' [e.g. 25]. We argue that, while tool use will certainly help, it is still important to improve base model capabilities in this regard, because oftentimes, even producing accurate inputs to a tool may require complex reasoning operations. Specifically, often we need to *copy* some part of the Transformer's input into a tool—if the base model struggles with robust copying, even this operation can be in peril.

Accordingly, we find it important to explain why decoder-only Transformers do not perform well when it comes to such problems—not just as an intellectual endeavour, but also to help guide further practical improvements. While many works have studied the computational capabilities of Transformers [22, 18], they often make assumptions which do not correspond to present practical limitations, such as infinite floating-point precision or 'hard attention', making their conclusions less directly practically applicable.

In this work, we take a different approach and study what information *can* be contained in the representation of the last token at the last layer, as this is ultimately the information that will be used for next-token prediction — the fundamental mechanism through which modern Transformer LLMs perform training and inference. In particular, we show that for certain distinct sequences, their last-token representations can become arbitrarily close to each other. This leads to a *representational collapse*, exacerbated by the lower-precision floating point types typically used by modern LLM stacks. As a result Transformers incorrectly produce the same tokens on these sequence pairs — see Figure 1 (a).

Furthermore, we reveal that the computation graph employed by decoder-only Transformers, with its unidirectional causal mask, contributes to the observed representational collapse. This unidirectional flow of information, converging at the final token, is in fact likely to lead to a loss of information due to *over-squashing*, an effect that is well studied in graph neural networks (GNNs) [2, 29, 8, 3, 11], and related to vanishing gradients [14, 5, 13]. We hope that this result will be of independent interest to the GNN community, as a practical application of over-squashing results at scale. Finally, we provide supporting empirical evidence that these issues are likely of practical interest, and propose simple solutions—directly stemming from our theoretical study—to help alleviate them.

In summary, our paper provides the following contributions:

- Theoretical analysis of decoder-only Transformer limitations: we formalise the concepts of 'representational collapse' (Section 4) and 'over-squashing' (Section 5) in the context of Transformer-based architectures.

- Impact of floating point precision: we explore how low floating-point precision exacerbate the identified theoretical issues, causing them to manifest even in relatively short input sequences.

- Empirical validation of theoretical analysis: our theoretical findings are supported by real-world experiments conducted on contemporary LLMs, demonstrating practical implications of the limitations we identified.

## 2  Background

In this work, we study a class of Transformers which we believe forms the basis for a large number of current LLMs. We let $\mathbf{Q}, \mathbf{K}, \mathbf{V} \in \mathbb{R}^{n \times d}$ be the query, key, and value matrices respectively on $n$ tokens and $d$ dimensions. We denote with $\mathbf{q}_i, \mathbf{k}_i, \mathbf{v}_i \in \mathbb{R}^d$ the $d$-dimensional query, key, and value vectors of the $i$-th token. We let $\mathbf{p}_{ij} \in \mathbb{R}^{2e}$ be the $2e$-dimensional positional encoding information between tokens $i$ and $j$. We focus on the case in which the positional encodings are *bounded*, which is the case for the large majority of positional encodings used in practice [26, 30]. The Transformer model we consider computes the values, for a single head, of the $i$-th token at the $\ell$-th Transformer layer $\mathbf{v}_i^{(\ell)}$ as[2]

$$\mathbf{z}_i^{(\ell)} = \sum_{j \leqslant i} \alpha_{ij}^{(\ell)} \, \mathrm{norm}_1^{(\ell)} \left( \mathbf{v}_i^{(\ell)} \right) + \mathbf{v}_i^{(\ell)}, \text{with } \alpha_{ij}^{(\ell)} = \frac{\exp\left( k\left( \mathbf{q}_i^{(\ell)}, \mathbf{k}_j^{(\ell)}, \mathbf{p}_{ij} \right) \right)}{\sum_{w \leqslant i} \exp\left( k\left( \mathbf{q}_i^{(\ell)}, \mathbf{k}_w^{(\ell)}, \mathbf{p}_{iw} \right) \right)}$$

$$\mathbf{v}_i^{(\ell+1)} = \boldsymbol{\psi}^{(\ell)} \left( \mathrm{norm}_2^{(\ell)} \left( \mathbf{z}_i^{(\ell)} \right) \right) + \mathbf{z}_i^{(\ell)}$$

for a function $k : \mathbb{R}^d \times \mathbb{R}^d \times \mathbb{R}^{2e} \to \mathbb{R}$ mapping queries, key, and positional encoding information to a scalar value, an MLP $\boldsymbol{\psi} : \mathbb{R}^d \to \mathbb{R}^d$, and normalization functions at the $\ell$-th layer $\mathrm{norm}_1^{(\ell)}$ and $\mathrm{norm}_2^{(\ell)}$. This specific interleaving of components is often referred to as a Pre-LN Transformer [34]. We can view the output of the $\ell$-th layer of a Transformer as a sequence of $d$-dimensional vectors $\mathbf{v}^{(\ell)} = (\mathbf{v}_1^{(\ell)}, \ldots, \mathbf{v}_n^{(\ell)})$. Importantly, due to the causal attention mechanism, the vector $\mathbf{v}_j^{(\ell)}$, will only depend on elements $\mathbf{v}_i^{(\ell-1)}$ for $i \leqslant j$. We can group the attention weights into an *attention matrix* at the $\ell$-th layer which we define element-wise as $\mathbf{\Lambda}_{ij}^{(\ell)} = \alpha_{ij}^{(\ell)}$. This is a row-stochastic triangular matrix that can also be interpreted as a probabilistic directed graph. After the last transformer block a normalization is applied to the token representations:

$$\mathbf{y}_i = \mathrm{norm}_3 \left( \mathbf{v}_i^{(L)} \right)$$

We note that the next-token prediction usually depends purely on $\mathbf{y}_n$—the final representation of the last token.

**Existing theory on Transformers.**   The theoretical representational capacity of Transformers has become a popular area of study, providing interesting results on what classes of problems they are able to model. For instance, it has been pointed out that Transformers are not Turing-complete, but one can apply modifications which make Transformers Turing-complete under certain assumptions [6]. Works have also shown that Transformers using 'hard attention' which replaces softmax with one-hot vectors alongside the use of infinite precision makes Transformers Turing-complete [22]. This contrasts with our work, which focuses on the more standard setting of Transformers using soft-attention and finite precision, and shows the limitations imposed by it.

Works have also tried to study transformers capabilities through the lense of formal languages, such as Weiss et al. [33], which develops a computational model of what transformers can represent in an analogous way to how Recurrent Neural Networks are associated with finite automata, and then derive an implementable programming language that represents

---

[2]Note that we rely on an abuse of notation. We ignore the linear projections used to compute the value $\mathbf{v}_i^{(l)}$ from the output of layer below $l - 1$. This will not change our derivations, but would otherwise make notations more cumbersome.

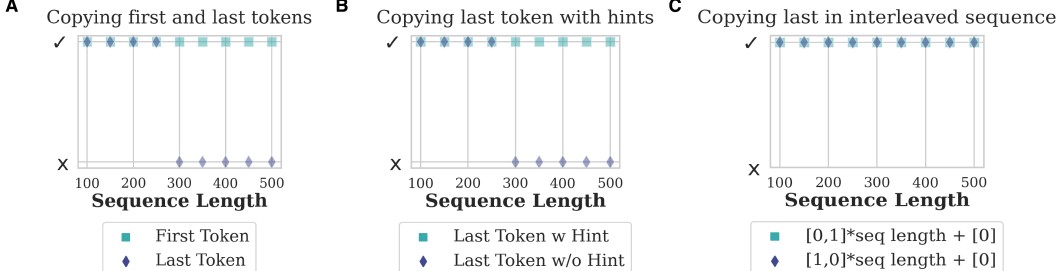

Figure 2: Results on simple copying tasks. (a). Gemini was prompted to predict the last token (diamond) of a sequences '1...10' or the first token (square) of a sequence '01...1'. (b). Same as (a) but with hints (see 3.2 for details) (c). Same as (a) but the sequences have interleaved 0s and 1s. See C.1 for extra details

that model. Following that, Delétang et al. [7] place transformers within the Chomsky Hierarchy, showing that they are quite limited and cannot learn the decision problem for simple languages, which prompted authors to show that Transformer LLMs can perform substantially better if they generate a number of decoding tokens linear in the problem input size, through scratch-pad, Chain-of-Thought (CoT) or similar [18]. Finally, Peng et al. [21] show that the Transformer block with finite precision is fundamentally limited in its ability to represent compositional functions and solve simple problems that require it. Our work similarly analyses the Transformer's inability to solve simple computational tasks, and proves that even with techniques like Chain-of-Thought that inability persists as it is inherent to the combination of architecture, next-token prediction, and limited floating point precision.

**Decay in attention mechanisms.** Works have also studied the limitations of self-attention by showing that it can reach pathological states that limit what transformers are able to learn. For instance, it has been show how a great reduction in the attention entropy can lead to unstable training if occurring early, but even when occurring later in training it can still lead to significantly lower performance [35]. Further, it has been shown that specific tokens can strongly concentrate attention, leading to transformers being unable to learn to process simple languages, like PARITY and DYCK [12]. Our work will similarly focus on showing how Transformers end-up effectively ignoring many tokens in their input which leads them to fail to solve simple computational problems, studying such a phenomenon by directly analysing the representational capacity.

**Over-squashing.** Graph neural networks (GNNs) are neural networks designed to operate over graph structures. Importantly, Transformers, may be seen as types of attention-based GNNs operating over specific types of graphs. The difficulties of propagating information over a graph have been thoroughly analysed, with a notable phenomenon being that of *over-squashing* [2, 29, 8, 3]. Over-squashing refers to the fact that propagating information over certain graphs that exhbit 'bottlenecks' is likely to induce a 'squashing' of information. This can be made more precise by studying this effect via the notion of a *commute time* [11] — the expected number of steps that a random walk takes to travel from a node to another node and back. Information travelling between nodes with higher commute time will be squashed more.

A common way to measure over-squashing is by looking at how *sensitive* the representation $\mathbf{x}_v^{(L)}$ of a node $v$ after $L$ GNN layers is to the initial representation $\mathbf{x}_u^{(0)}$ of another node $u$. In particular, the partial derivative $\partial \mathbf{x}_v^{(L)}/\partial \mathbf{x}_u^{(0)}$ may be shown to decay, especially for nodes with high commute times between them. Our work may be seen as acting as a bridge between the well-studied phenomenon of over-squashing in GNNs and the loss of information we analyse in decoder-only Transformers specifically for language tasks. Note that this type of derivation is typical in the study of *vanishing gradients* for recurrent models as well [14, 5, 20, 13].

## 3 Motivating Examples

This section presents a series of experiments focused on copying and counting tasks. These experiments reveal surprising failure cases in modern decoder-only Transformer architectures,

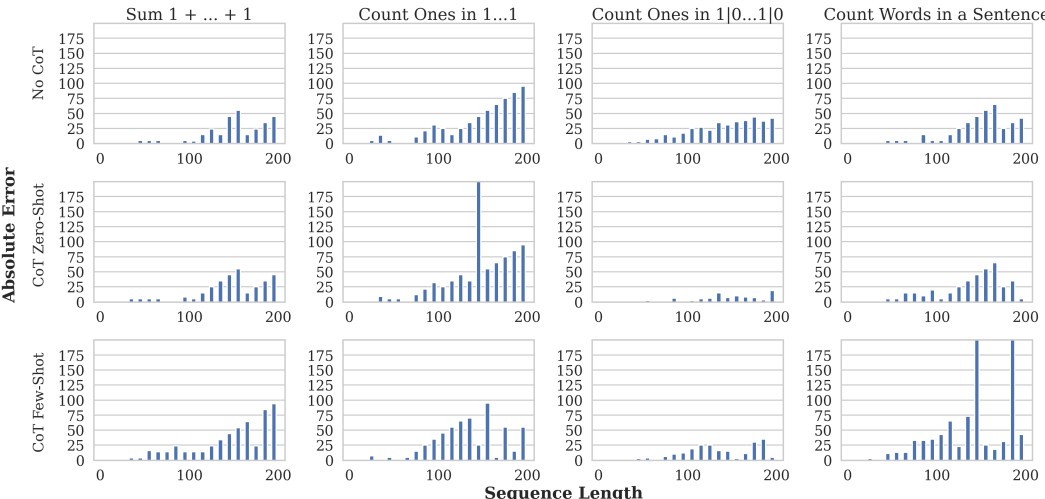

Figure 3: Gemini 1.5 being prompted to sum $1 + \cdots + 1$ (Column 1), Count the number of ones in a sequence of 1s (Column 2), Count the number of ones in a sequence of ones and zeroes (the sequence is a Bernoulli sequence with probability of sampling a one being 0.7) (Column 3), and to counter the number of times a word appears in a sentence (Column 4).

providing concrete evidence that motivates the theoretical analysis presented in the following sections.

We start by providing motivating examples that show surprisingly simple failure cases of frontier LLMs specifically on copying (Section 3.1) and counting (Section 3.2) tasks. By copying we specifically mean tasks that involve the 'copying' or 'recalling' of a single or multiple tokens from the prompt. Instead, by *counting*, we mean the task of counting how many times a specific token appears in a sequence. We focus our evaluation on Gemini 1.5 [10] as our frontier LLM (referred as Gemini) and later analyse the internal representations of the open-sourced Gemma model [27]. The goal is to showcase intriguing failure cases which will motivate our signal propagation analysis.

## 3.1 Copying

In this Section, we present surprising results on simple *copying* tasks. In particular, we focus on tasks that involve the copying of a *single* token — i.e. what is the token occurring at a particular position? The copy of a single token is in principle the most straightforward type of copying task, but still requires the LLM to accurately identify the token based on a prompt and to then propagate its information correctly.

Importantly, we study cases in which the LLM is prompted to copy tokens either at the *start* or at the *end* of a sequence. We avoid tasks that involve the copy of tokens at the '*n*-th' position as most frontier LLMs do not have absolute positional information, making it very challenging for them to solve tasks that require absolute position. We focus on tasks that involve sequences of 'zeros' and 'ones' growing in length with specific patterns.

In Figure 2 (a), we prompt Gemini to copy the last element of a sequence '1 . . . 10' or the first element of a sequence '01 . . . 1'. The answer for both is zero, but we progressively grow the number of ones. We observe how the task seems considerably easier when asked to return the first rather than the last element. Surprisingly, already at a sequence length of only 300 elements, Gemini incorrectly starts to output 'one' when trying to copy the last element. In Figure 2 (b), we show that providing hints in the form of: " *Hint* It's not necessarily a 1, check carefully", helps significantly with the performance. Finally, in Figure 2 (c), we show that replacing the constant sequence of ones with alternating ones and zeros seems to also help. We refer to the Appendix (Section C.1) for further details on the experiments.

These three motivating experiments seem to point towards a type of vanishing of information, caused by the growing number of ones dominating the sequence. Interestingly, such a

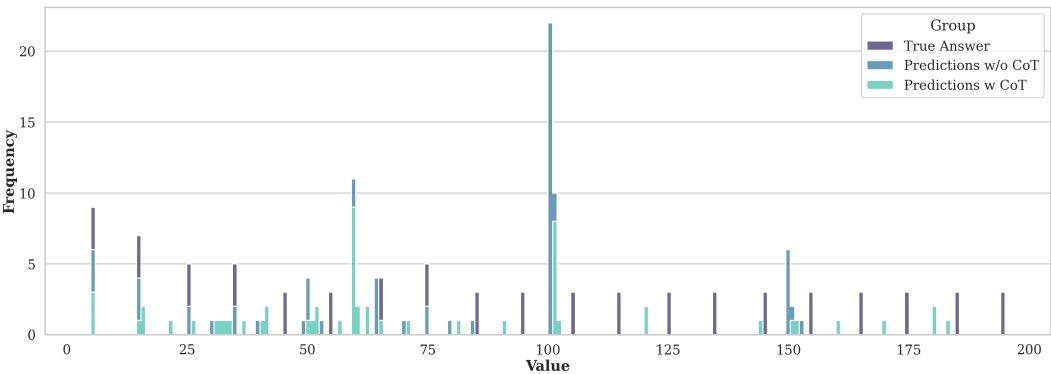

Figure 4: Frequency of different outputted values for Gemini 1.5 for the counting tasks. The large density at 100 suggests that Gemini is likely not counting, but instead possibly performing some crude form of subitising.

vanishing of information effect (a) seems to depend on the position in the sequence, (b) seems to be affected by the prompting, and (c) by the items that make up the sequence. We will later argue how all three of such observations can explained by our theoretical analysis.

## 3.2 Counting

We now turn our attention to *counting* problems, i.e. tasks of the form — given a specific sequence, how many times does a particular token appear? Such problems are related to copying in the sense that they also require careful consideration of individual tokens in the sequence as ignoring even a single token may potentially lead to an incorrect output.

We consider four different tasks: (i) Summing $1 + \cdots + 1$, (ii) Counting the number of ones in a sequence of ones, (iii) Counting the number of ones in a sequence of ones and zeros, with ones being sampled with 70% probability, and (iv) Counting the number of times a specific word appears in a sentence. We consider predictions of an LLM which (1) Is instructed to only output the answer, (2) Is prompted to break down the problem (CoT-no-shot), and (3) Is prompted to break down the problem with few-shot in-context examples (CoT-few-shot). We refer to the Appendix (Section C.1) for a more detailed description of the tasks.

Results are presented in Figure 3. It is clear that the performance rapidly deteriorates with the sequence length. It is also interesting to see that the error seems to increase with the sequence very rapidly. For instance in task (i), the LLM is quite likely to predict the value of '100' once the sequence reaches a size around or larger than 100. Such an observation provides motivating evidence for the argument that Transformers may not be in fact mechanically counting but rather perform a type of crude subitising. This explains why arguably 'common' numbers such as 100 are much more likely to be outputted by the LLM and why in tasks such as (i) and (ii) the values near 100 have relatively lower error. This does not happen in task (iii) as the response should actually be around 70% of the sequence length due to the sequence sampling procedure, explaining why the absolute error actually seems to increase around a sequence length of 100. Figure 4 further showcases this issue, more clearly showing how 100 is by far the most common response.

## 4 Representational Collapse

We start our theoretical analysis by showcasing a type of loss of information which we call *representational collapse*. More precisely, we show that under certain conditions, we can find distinct sequences such that their final representations of the last token at the last layer *become arbitrarily close* as the sequence length increases. As Transformer models operate over finite machine precision, this points to a fundamental representational incapacity of Transformers to distinguish certain prompts if the sequence is long enough.

The key intuition is that if two sequences are similar everywhere except at the last token, as the sequences get larger, their final representations will become closer and closer until they

reach a critical point which is below floating point precision. In other words, solving certain tasks would require infinite floating point precision. We will later show how this phenomenon is not only theoretical, but also occurs in practice on sequences of reasonable length. In the Appendix (Section 4), we relate representational collapse to the $L_1$ distance – or total variation – between the softmax distributions of the two sequences. We start by presenting a result that shows that the $L_1$ difference tends to 0 as the sequence length grows, under some assumption on the sequences. We point to the Appendix (Lemma B.2) for the complete statement.

**Lemma 4.1** (Informal). *Consider two sequences* $\mathbf{x}, \mathbf{x}^* \in \mathbb{R}^n$ *such that* $\lim_{n \to \infty} |\mathbf{x}_n - \mathbf{x}_n^*| = 0$. *Then, the* $L_1$ *difference of their softmax tends to* 0.

We now show, using Lemma 4.1, that we can find distinct sequences that will have arbitrarily close final representations. In particular, as language models often operate in low floating regimes, i.e. `bf16`, this can practically become catastrophic. The result is summarised in Theorem 4.2, which describes what we call representational collapse in this work. The complete statement is reported in the Appendix (Theorem B.3).

**Theorem 4.2** (Representational Collapse – informal). *Let* $\mathbf{v}^{(0)} \in \mathbb{R}^{n \times d}$ *be a sequence and* $\mathbf{v}^{*(0)} \in \mathbb{R}^{(n+1) \times d}$ *be another sequence equal to* $\mathbf{v}^{(0)}$ *with the last token of* $\mathbf{v}^{(0)}$ *repeated. Assume that the positional encoding information decays to* 0 *with the distance. Then, their representations become arbitrarily close as n increases.*

Theorem 4.2 shows that it becomes increasingly challenging for a Transformer to distinguish two sequences that only differ via a repeated last token. We note that the repetition of the last token is a technical consideration to show this direct representational collapse. As we will later show in Section 5.1, it is particularly problematic *in general* to depend on the last token due to a type of topological 'squashing' present in decoder-only Transformers.

**Measuring representational collapse.** We report experiments showcasing representational collapse by measuring the internal representations of Gemma 7B [27]. For two sequences $\mathbf{v}^{(0)}$ and $\mathbf{v}^{*(0)}$ we report their difference in representation at the last layer $\left\| \mathbf{v}^{(L)} - \mathbf{v}^{*(L)} \right\|_\infty$ averaged out over each head, alongside the minimum and maximum over each head. Figure 5 shows the collapse occuring on (a) prompting the model to count the number of ones in a sequence of ones, with one having an additional one, and (b) prompting the model to count the number of ones for a sequences with digits sampled uniformly ending with either a single one or two ones. The repeated digits seem to make the collapse occur much sooner with a sequence length of around 50 being near machine precision, while varying the digits seems to delay such a collapse, but a downward trend is maintained with respect to the sequence length.

**Quantisation and Tokenisation.** A common technique used to speedup the inference of an LLM is that of *quantisation*, a process that constructs an approximate version of an LLM that operates over lower precision datatypes. This helps drastically improve the inference speed of LLMs as modern accelerators produce significantly more FLOPs over lower precision datatypes. Of course quantisation usually comes at a cost. Our theoretical analysis points towards a potentially catastrophic loss in representation due to quantisation. In particular, a lower machine precision will mean that the convergence of representations in Theorem 4.2 will occur much sooner, and consequently the LLM will not be able to distinguish even shorter sequences.

In practice, the direct application of theoretical results is made more complicated due to tokenisation. In particular, a sequence of repeated tokens '11111' for instance may not be necessarily tokenised into 5 distinct '1' tokens. In principle, this should help alleviate the direct collapse of the representations. Tokenisation in general makes it more of a challenge to study such phenomena as it adds an additional layer of complexity to the experimental analysis. In our experiments, we took tokenisation into consideration and attempted to mitigate its effects.

**A simple solution to representational collapse.** An important consequence of Theorem 4.2 is that *it is challenging for a Transformer to deal with a long sequence of repeated tokens*. A practical solution is to this issue is to introduce additional tokens throughout the sequence to help keep the representations distant. We provide direct evidence of this in Figure 5 (c,d), where we prompt the model on a simple copying task of a long string of ones. While the

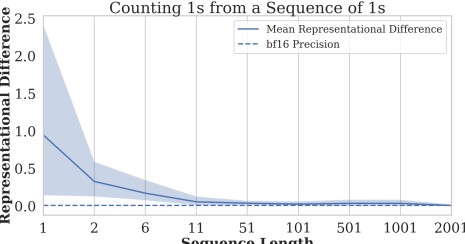

(a) "How many ones are in the following sequences?" Followed by a sequence of ones.

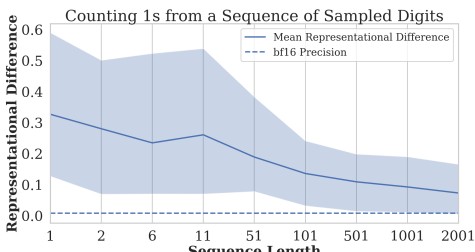

(b) "How many ones are in the following sequences?" Followed by sampled digits.

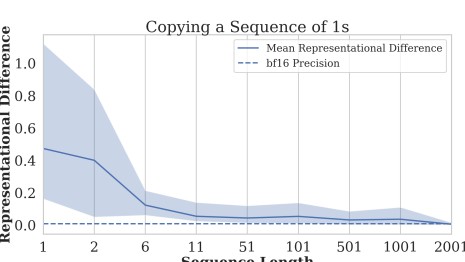

(c) "Can you copy the following number?" Followed by a sequence of ones.

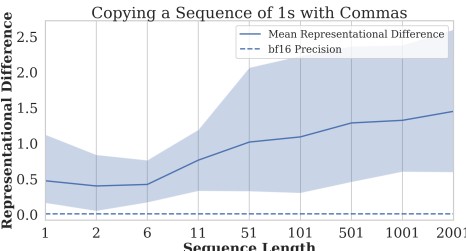

(d) "Can you copy the following number?" Followed by a sequence of ones with commas.

Figure 5: Representational collapse for counting (a, b) and copying (c, d) tasks.

representations collapse for the sequence of ones (c), adding commas every third digit (d) helps to keep the representations well-separated.

## 5 Over-squashing in Language Tasks

In this Section, we study a more general phenomenon related to representational collapse—over-squashing. In particular, we are interested in analysing how information from the input sequence affects the information contained within the representation of the last token in the final layer—the representation ultimately used for next-token prediction. For this reason, we study the quantity $\partial \mathbf{y}_n / \partial \mathbf{v}_i^{(0)}$ which measures how sensitive is the final token to an input token at position $i$.

In graph neural network theory, the decay of such a partial derivative is often associated with the 'squashing' of information, leading to the phenomenon of *over-squashing* [29, 8, 11], a problem related to the well-known vanishing gradients problem in RNNs [14, 5, 11]. The over-squashing analysis we carry out in this work is particularly challenging due to the flexible nature of the attention mechanism and the many components that are part of decoder-only Transformers. Consequently, we make two simplifying assumptions in our analysis: (i) We summarise the effect of layer normalisation via a constant $\beta_i$ for the $i$-th layer norm component, and (ii) the attention weights are treated as independent of the input. Such simplifications are not strictly necessary for our analysis, but they greatly simplify the resulting bound we derive and do not detract from the two key takeaways: **(1) the sensitivity to an input token depends on its position in the sequence and (2) the sensitivity to an input token depends on the attention weights**. The result is summarised in Theorem 5.1. The full statement is reported in the Appendix (Theorem B.5).

**Theorem 5.1** (Over-squashing in Transformers). *Consider an input sequence $\mathbf{v}_1^{(0)}, \ldots, \mathbf{v}_n^{(0)}$. Let $C > 0$ be some constant and $\bar{\alpha}_{i,j}^{(\ell)} = \frac{\alpha_{i,j}^{(\ell)}}{\beta_2} + \delta_{i,j}$, then:*

$$\left\| \frac{\partial \mathbf{y}_n}{\partial \mathbf{v}_i^{(0)}} \right\| \leqslant C \sum_{k_1 \geqslant i} \cdots \sum_{k_L \geqslant k_{L-1}} \bar{\alpha}_{n,k_L}^{(L-1)} \prod_{\ell=2}^{L-1} \bar{\alpha}_{k_\ell, k_{\ell-1}}^{(\ell-1)} \bar{\alpha}_{k_1, i}^{(0)} \tag{1}$$

Theorem 5.1 provides intuition on how information propagates in a decoder-only Transformer. In particular, there is a topological aspect present in the bound which is directly controlled by the attention mechanism. More concretely, the sensitivity depends on the sum of the weighted paths between the token $i$ at the input and the final layer. *In other words, for tokens coming sooner in the sequence, there will be more opportunity for their information to be preserved.* This is clear for instance for the last token, which will only be preserved by attention mechanism if the attention $n \to n$ is large at every layer $L$, i.e. there is only one path. The paths instead grow very quickly for tokens coming sooner in the sequence. A related observation, in terms of path counting, was also made for deep RNNs [13]. We note that such a bound explains the better performance when copying elements at the start of the sequence in Figure 2 (a), why hints help in Figure 2 (b), and why repeating the final elements within the sequence also helps in Figure 2 (c).

This analysis leads to an interesting limiting case described in Proposition 5.2, that shows a type of exponential vanishing that can occur in some degenerate cases in which $\mathbf{y}_n$ depends only on the starting input token $\mathbf{v}_1^{(0)}$. Fortunately, there are many mechanisms which prevent this from happening, but believe this to be an interesting limiting case which is a direct consequence of the topology of the causal attention mechanism. Further, it provides an interesting connection between the spectral theory of directed graphs and causal attention mechanisms. We report the formal statement in the Appendix (Proposition B.8).

**Proposition 5.2** (Informal)**.** *Under certain assumptions on the effect of the normalisation and on the attention weights, in the limit of layers $L \to \infty$ the output representation will only depend on the first input token.*

**U-shape effect.** Theorem 5.1 in part also helps to explain the empirically observed *U-shape effect*—the observation that LLMs seem to perform better at retrieval tasks when the information to be retrieved is located either near the start or the end of the sequence. In fact, due to the topology of the causal mechanism, we find from Theorem 5.1 that tokens at the start of the sequence have more opportunity for the information to be maintained at the end. The final tokens being also easier instead can be explained from the recency bias that is learnt by the attention mechanism during training. In auto-regressive next-token prediction, it is in fact reasonable to assume that tokens that are closer to the end will be more important and this is likely a bias that is learnt during training by the LLM.

## 6 Counting

We finally highlight another representational problem that arises specifically in counting problems. Our analysis points to a fundamental difficulty that emerges from the normalisation of the softmax. In particular, the normalisation of the softmax makes it hard for a model to take into account the *length* of a sequence. This is exacerbated by the fact that positional encodings are often normalised and thus relative, meaning that they also do not hold absolute positional information. Intuitively, counting is a problem that requires some notion of 'unboundedness' of the representations, whilst the normalisations used inside a Transformer work against this.

We start by showing that without causal masking and positional embeddings, a Transformer is immediately unable to count the number of tokens in a sequence, highlighting a pathological issue which stems directly from the softmax normalisation. We note that similar issues have been already pointed out [e.g. 22]. We show the result in Proposition 6.1 and report the full statement in the Appendix (Proposition B.9).

**Proposition 6.1.** *A Transformer without positional encodings and a causal attention mechanism is immediately unable to count.*

While causal mechanisms and positional encodings help to break such representational issues, they break the permutation invariance of the Transformer, meaning that the representations will be heavily miss-aligned with the task, something which has been shown to hinder performance [9]. As permutations grow factorially with sequence length, this makes it practically very challenging for a decoder-only Transformer to learn such a property simply from the data. This explains the extreme incapacity of counting highlighted in Section

3. Further, as a corollary of Theorem 4.2, we have that even if a model would be able to generalise perfectly, the problem of representational collapse points to an impossibility result in counting regardless. The result is summarised in Corollary 6.2, with the full statement in the Appendix (Corollary B.10).

**Corollary 6.2** (Informal)**.** *Counting in certain situations becomes impossible due to representational collapse and finite floating point precision.*

Corollary 6.2 shows how our main result on representational collapse points to practical issues when it comes to certain styles of prompts. When paired with low floating point arithmetic precision, representation collapse becomes problematic.

## 7   Conclusion and Future Work

In this work, we first presented surprising failure cases of LLMs on simple copying and counting tasks. We then discussed how such failure cases can be explained by studying what *can* be contained inside the representation $\mathbf{y}_n$ and in particular how information may be lost. This lead to the unvealing of two phenomena : representational collapse and over-squashing. We showed how we can measure these phenomena in practice and proposed simple solutions to help alleviate such information loss.

We believe that this work uncovers an interesting framework which can be used to study failure cases of Transformers and LLMs more generally. We believe that our analysis could be extended in many practical different directions, for instance by understanding how to directly measure over-squashing or how to best use this newly-found understanding to improve current Transformer models. In our work, we focused on pointing out information-propagation issues in Transformer-based architectures, but we hope that the findings may help better understand and improve language models available today.

## 8   Acknowledgements

We would like to thank Wojciech Marian Czarnecki (Google DeepMind), Simon Osindero (Google DeepMind), and Timothy Nguyen (Google DeepMind) for the valuable comments and suggestions regarding this work. We also thank Constantin Kogler (University of Oxford) for providing useful insights on the theoretical aspects of the work.

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

## A   Broader impact

This paper presents work whose goal is to advance the field of Machine Learning. There are many potential societal consequences of our work, none which we feel must be specifically highlighted here. In particular the limitations highlighted in the work can pose some issue in terms of reliability of LLMs, so highlighting these issues can help fixing them.

## B   Proofs

We provide formal statements and proofs for the results shown in the main text. We follow the order in which they are presented in the main text. In Section B.1, we present the proofs on representational collapse (Section 4), in Section B.2 the proofs on over-squashing (Section 5) over-squashing, and finally in Section B.3 the proofs on counting (Section 6).

### B.1   Representational Collapse

We start by showing that adding a new element to a sequence results in the softmax value of a specific token to decrease. In particular, we consider the case in which the tokens are bounded and show that we can use this to construct an upper bound on the softmax value for any token.

**Lemma B.1.** *Consider a vector* $\mathbf{a} \in \mathbb{R}^{n-1}$ *and two scalars* $b, c \in \mathbb{R}$. *Let* $\mathbf{x} = [\mathbf{a} \ c]^T \in \mathbb{R}^n$ *and* $\mathbf{x}^* = [\mathbf{a} \ b \ c]^T \in \mathbb{R}^{n+1}$ *with all entries bounded. Then,* $\mathrm{softmax}\,(\mathbf{x})_n > \mathrm{softmax}\,(\mathbf{x}^*)_{n+1}$. *Moreover for any* $p > 0$ *we can find large enough* $n \in \mathbb{N}^+$ *such that* $|\,\mathrm{softmax}\,(\mathbf{x})_n - \mathrm{softmax}\,(\mathbf{x}^*)_{n+1}\,| < p$.

*Proof.* We directly compute:

$$\mathrm{softmax}(\mathbf{x})_n = \frac{\exp(c)}{\sum_{k=1}^{n-1} \exp(\mathbf{a}_k) + \exp(c)}$$

$$\mathrm{softmax}(\mathbf{x}^*)_{n+1} = \frac{\exp(c)}{\sum_{k=1}^{n-1} \exp(\mathbf{a}_k) + \exp(b) + \exp(c)}$$

As we assume that the entries are bounded, we have that $\sum_{j=1}^{n-1} \exp(\mathbf{a}_j) + \exp(c) < \sum_{j=1}^{n-1} \exp(\mathbf{a}_j) + \exp(b) + \exp(c)$, therefore $\mathrm{softmax}(\mathbf{x})_n > \mathrm{softmax}(\mathbf{x}^*)_{n+1}$.

For the second part of the statement, we compute:

$$
\begin{aligned}
|\mathrm{softmax}(\mathbf{x})_n - \mathrm{softmax}(\mathbf{x}^*)_{n+1}| &= \left| \frac{\exp(c)}{\sum_{k=1}^{n-1} \exp(\mathbf{a}_k) + \exp(c)} - \frac{\exp(c)}{\sum_{k=1}^{n-1} \exp(\mathbf{a}_k) + \exp(b) + \exp(c)} \right| \\
&\leqslant \left| \frac{\exp(c)}{\sum_{k=1}^{n-1} \exp(\mathbf{a}_k) + \exp(c)} \right| + \left| \frac{\exp(c)}{\sum_{k=1}^{n-1} \exp(\mathbf{a}_k) + \exp(b) + \exp(c)} \right| \\
&< p
\end{aligned}
$$

for some $p > 0$, as in the last step the summands tend to 0 as $n \to \infty$. We therefore have that $|\mathrm{softmax}(\mathbf{x})_n - \mathrm{softmax}(\mathbf{x}^*)_{n+1}| \to 0$ as $n \to \infty$ and for large enough $n$ $|\mathrm{softmax}(\mathbf{x})_n - \mathrm{softmax}(\mathbf{x}^*)_{n+1}| < p$ for any $p > 0$ due to the previous statement. $\square$

**Total variation.**   We now study a quantity known as the *total variation* between two distributions of interest. Given two categorical distributions $\mu, \nu$ supported on the same space, we define their total variation $\delta(\mu, \nu)$ — or equivalently $L_1$ norm $\|\mu - \nu\|_1$, as:

$$\delta(\mu, \nu) = \sum_x |\mu(x) - \nu(x)|. \tag{2}$$

The total variation is a distance between probabilty distributions. We note that oftentimes the quantity above is strictly called the $L_1$ norm, while $1/2$ of such a quantity the total variation. For the scope of our work, the factor of $1/2$ is not important so we ignore it and use the two terms synonymously. Interestingly, the total variation is intimately related to the KL-divergence, pointing towards potential connections to information theory. We leave such a connection to future work.

We now study how the total variation between two softmax distributions behaves in the limit of the sequence length. In particular, we show that if the positional encoding information decays to 0 with the sequence length, then the total variation between the two sequences goes to 0 with $n$. Such a result is presented in Lemma B.2

**Lemma B.2.** *Consider two sequences $\mathbf{x}, \mathbf{x}^* \in \mathbb{R}^n$ such that $\lim_{n \to \infty} |\mathbf{x}_n - \mathbf{x}_n^*| = 0$, with $\mathbf{x}_i, \mathbf{x}_i^*$ bounded. Let $\mathbf{y}, \mathbf{y}^* \in \mathbb{R}^n$ be the softmax of $\mathbf{x}$ and $\mathbf{x}^*$ respectively. Then, as $n \to \infty$ the total variation tends to 0, i.e. $\lim_{n \to \infty} \delta(\mathbf{y}, \mathbf{y}^*) = 0$.*

*Proof.* Let $Z = \sum_{i=1}^n e_i^{\mathbf{x}}$ and $Z^* = \sum_{i=1}^n e_i^{\mathbf{x}^*}$ be the partition functions for $\mathbf{x}$ and $\mathbf{x}^*$, respectively. We start by bounding the quantity $|Z - Z^*|$. In particular, let $\epsilon > 0$, we first claim:

$$|Z - Z^*| \leqslant \sum_{i=1}^n \left| e^{\mathbf{x}_i} - e^{\mathbf{x}_i^*} \right| \leqslant \epsilon \min(Z, Z^*). \tag{3}$$

Consider some $n_0 \geqslant 1$ such that $\left| 1 - e^{\mathbf{x}_i^* - \mathbf{x}_i} \right| \leqslant \epsilon/2$. We note that this always possible as $|\mathbf{x}_i^* - \mathbf{x}_i| \to 0$ by assumption. We compute:

$$
\begin{aligned}
\sum_{i=1}^n \left| e^{\mathbf{x}_i} - e^{\mathbf{x}_i^*} \right| &= \sum_{i=1}^{n_0-1} \left| e^{\mathbf{x}_i} - e^{\mathbf{x}_i^*} \right| + \sum_{i=n_0}^n \left| e^{\mathbf{x}_i} - e^{\mathbf{x}_i^*} \right| \\
&= \sum_{i=1}^{n_0-1} \left| e^{\mathbf{x}_i} - e^{\mathbf{x}_i^*} \right| + \sum_{i=n_0}^n e^{\mathbf{x}_i} \left| 1 - e^{\mathbf{x}_i^* - \mathbf{x}_i} \right| \\
&\leqslant \sum_{i=1}^{n_0-1} \left| e^{\mathbf{x}_i} - e^{\mathbf{x}_i^*} \right| + \frac{\epsilon}{2} \sum_{i=n_0}^n e^{\mathbf{x}_i} \\
&\leqslant \sum_{i=1}^{n_0-1} \left| e^{\mathbf{x}_i} - e^{\mathbf{x}_i^*} \right| + \frac{\epsilon}{2} Z \\
&\leqslant \epsilon Z
\end{aligned}
$$

Where the last step comes from the observation that the first sum is fixed and $Z$ is unbounded with $n$. Therefore, for $n$ large enough, we can also bound the left summand by $Z\epsilon/2$. The same argument also holds when bounding with $Z^*$ instead of $Z$, leading to the claim in Equation 3.

We now proceed with the following computation:

$$\delta(\mathbf{y}, \mathbf{y}^*) = \sum_{i=1}^{n} \left| \frac{e^{\mathbf{x}_i}}{Z} - \frac{e^{\mathbf{x}_i^*}}{Z^*} \right|$$

$$= \sum_{i=1}^{n} \left| \frac{Z^* e^{\mathbf{x}_i} - Z e^{\mathbf{x}_i^*}}{ZZ^*} \right|$$

$$= \sum_{i=1}^{n} \left| \frac{Z^* e^{\mathbf{x}_i} - Z e^{\mathbf{x}_i} + Z e^{\mathbf{x}_i} - Z e^{\mathbf{x}_i^*}}{ZZ^*} \right|$$

$$\leqslant \sum_{i=1}^{n} \left| \frac{Z^* e^{\mathbf{x}_i} - Z e^{\mathbf{x}_i}}{ZZ^*} \right| + \left| \frac{Z e^{\mathbf{x}_i} - Z e^{\mathbf{x}_i^*}}{ZZ^*} \right|$$

$$= \sum_{i=1}^{n} \frac{|Z^* - Z| \, e^{\mathbf{x}_i}}{ZZ^*} + \frac{Z |e^{\mathbf{x}_i} - e^{\mathbf{x}_i^*}|}{ZZ^*}$$

$$= \frac{|Z^* - Z|}{ZZ^*} Z + \sum_{i=1}^{n} \frac{|e^{\mathbf{x}_i} - e^{\mathbf{x}_i^*}|}{Z^*}$$

$$\leqslant \frac{\epsilon \min(Z, Z^*)}{Z^*} + \frac{\epsilon \min(Z, Z^*)}{Z^*}$$

$$\leqslant 2\epsilon$$

which concludes the proof. $\qquad\square$

We are now ready to show the main result on representational collapse. In particular, we show that given two sequences of length $n$ and $n+1$ where the second sequence is the same as the first with a final repeated token, their representations become arbitrarily close. Importantly, we require that the information from the positional encodings decays to 0 as the distance grows between tokens. The final token repetition is important as otherwise the residual connection in the Transformer would not make the representations necessarily converge.

**Theorem B.3** (Representational Collapse). *Let $\mathbf{x} \in \mathbb{R}^{n-1 \times d}$ be an underlying growing token sequence. Let $\mathbf{v}^{(0)} = [\mathbf{v} \ \mathbf{v}_a]^T \in \mathbb{R}^{n \times d}$ and $\mathbf{v}^{*(0)} = [\mathbf{v} \ \mathbf{v}_a \ \mathbf{v}_a]^T \in \mathbb{R}^{n+1 \times d}$ be two sequences for a final repeated token $\mathbf{x}_a \in \mathbb{R}^d$, with all token representations bounded. Further, assume that the positional encodings decay with distance to $0$. Then, for large enough $n \in \mathbb{N}^+$, we have that the representations are under any $\epsilon$:*

$$||\mathbf{v}_n^{(L)} - \mathbf{v}_{n+1}^{*(L)}||_1 < \epsilon.$$

*Proof.* We note that since the sequences are identical up to the $n$-th element, it is sufficient to only check the representations of the final elements in both sequences. We therefore compare the $n$-th element of $\mathbf{z}^{(0)}$ with the $n+1$-th element of $\mathbf{z}^{*(0)}$:

$$\left\| \mathbf{z}_n^{(0)} - \mathbf{z}_{n+1}^{*(0)} \right\|_1 = \left\| \sum_{i<n} \alpha_{n,i}^{(0)} \mathbf{v}_i^{(0)} + \alpha_{n,n}^{(0)} \mathbf{v}_a^{(0)} - \left( \sum_{i<n} \alpha_{n+1,i}^{*(0)} \mathbf{v}_i^{(0)} + \left( \alpha_{n+1,n}^{*(0)} + \alpha_{n+1,n+1}^{*(0)} \right) \mathbf{v}_a^{(0)} \right) \right\|_1$$

$$= \left\| \sum_{i<n} \left( \alpha_{n,i}^{(0)} - \alpha_{n+1,i}^{*(0)} \right) \mathbf{v}_i^{(0)} + \left( \alpha_{n,n}^{(0)} - \alpha_{n+1,n}^{*(0)} - \alpha_{n+1,n+1}^{*(0)} \right) \mathbf{v}_a^{(0)} \right\|_1$$

$$\leqslant \sum_{i<n} \left| \alpha_{n,i}^{(0)} - \alpha_{n+1,i}^{*(0)} \right| + \left| \alpha_{n,n}^{(0)} - \alpha_{n+1,n}^{*(0)} - \alpha_{n+1,n+1}^{*(0)} \right|$$

$$\leqslant \sum_{i<n} \left| \alpha_{n,i}^{(0)} - \alpha_{n+1,i}^{*(0)} \right| + \left| \alpha_{n,n}^{(0)} - \alpha_{n+1,n}^{*(0)} \right| + \left| \alpha_{n+1,n+1}^{*(0)} \right|$$

$$= \delta \left( \alpha_{n,:}^{(0)}, \alpha_{n,:n}^{*(0)} \right) + \alpha_{n+1,n+1}^{*(0)} < \epsilon$$

We assume for simplicity that the values are unit norm. This is not crucial as otherwise one would equivalently just need to consider additional constant factors as we assume the token representations are bounded. We note that the term $\delta\left(\alpha_{n,:}^{(0)}, \alpha_{n,:n}^{*(0)}\right)$ goes to 0 with $n \to \infty$ thanks to Lemma B.2 and our assumptions on the positional encodings. Similarly, the term $\alpha_{n+1,n+1}^{*(0)}$ goes to 0 due to Lemma B.1. We have also used the fact that $\mathbf{v}_n^{(0)} = \mathbf{v}_{n+1}^{*(0)}$ by construction to ignore the residual connection. As $\mathbf{v}_i^{(\ell+1)} = \boldsymbol{\psi}^{(\ell)}\left(\text{norm}_2^{(\ell)}\left(\mathbf{z}_i^{(\ell)}\right)\right) + \mathbf{v}_i^{(\ell)}$, the sequences will have arbitrarily close final token representations when entering the next layer. The result then follows via a simple inductive approach on the layers. $\qquad\square$

We also highlight a negative decay result which highlights why the assumption on the positional encodings is important. In particular, we show that given two sequences $\mathbf{x} = (1\ 0\ 1\ 0\ldots)$ and $\mathbf{x}^* = (0\ 1\ 0\ 1\ldots)$, the total variation of the softmax does not decay to 0. This implies that there may be solutions to representational collapse depending on how the positional encodings are chosen.

**Proposition B.4.** *Consider two sequences* $\mathbf{x} = (1\ 0\ 1\ 0\ldots) \in \mathbb{R}^n$ *and* $\mathbf{x}^* = (0\ 1\ 0\ 1\ldots) \in \mathbb{R}^n$ *for $n$ even. Let $\mathbf{y}$ and $\mathbf{y}^*$ be the softmax of $\mathbf{x}$ and $\mathbf{x}^*$, respectively. Then the total variation $\delta(\mathbf{y}, \mathbf{y}^*)$ does not tend to 0 as $n \to \infty$.*

*Proof.* Let $Z = \sum_{i=1}^{n} e_i^{\mathbf{x}}$ and $Z^* = \sum_{i=1}^{n} e^{\mathbf{x}_i^*}$ be the partition functions for $\mathbf{x}$ and $\mathbf{x}^*$, respectively. We directly compute:

$$
\begin{aligned}
\lim_{n\to\infty} \delta(\mathbf{y}, \mathbf{y}^*) &= \lim_{n\to\infty} \sum_{i=1}^{n} \left| \frac{\mathbf{y}_i}{Z} - \frac{\mathbf{y}_i^*}{Z^*} \right| \\
&= \lim_{n\to\infty} \sum_{i=1}^{n} \left| \frac{\mathbf{y}_i}{\frac{n}{2}e + \frac{n}{2}} - \frac{\mathbf{y}_i^*}{\frac{n}{2}e + \frac{n}{2}} \right| \\
&= \lim_{n\to\infty} \sum_{i=1}^{n} \left| \frac{e-1}{\frac{n}{2}e + \frac{n}{2}} \right| \\
&= \lim_{n\to\infty} \frac{n(e-1)}{n\frac{e+1}{2}} \\
&= 2\frac{e-1}{e+1} > 0.
\end{aligned}
$$

$\qquad\square$

## B.2 Over-squashing

We now present our results on over-squashing. In our derivations, we assume that the attention coefficients are independent of the values and that we can summarise the effect of the layer norms via a constant factor. These assumptions are not necessary for the same derivation process to hold, but they greatly simplify the obtained bound and help more clearly point out the main takeaways.

**Theorem B.5** (Over-squashing in Transformers). *Consider an input sequence $\mathbf{v}_1^{(0)}, \ldots, \mathbf{v}_n^{(0)}$ (including CoT). Let $\sigma_{\boldsymbol{\psi}}$ be the maximal Lipschitz constant of any $\boldsymbol{\psi}^{(\ell)}$, and $\bar{\alpha}_{j,i}^{(\ell)} = \frac{1}{\beta^{(\ell)}}\left(\alpha_{j,i}^{(\ell)} + \delta_{j,i}\right)$ the normalized attention coefficient, then:*

$$
\left\| \frac{\partial \mathbf{y}_n}{\partial \mathbf{v}_i^{(0)}} \right\| \leqslant \sigma_{\boldsymbol{\psi}}^L \sum_{k_1 \geqslant i} \cdots \sum_{k_L \geqslant k_{L-1}} \bar{\alpha}_{n,k_L}^{(L-1)} \prod_{\ell=2}^{L-1} \bar{\alpha}_{k_\ell, k_{\ell-1}}^{(\ell-1)} \bar{\alpha}_{k_1, i}^{(0)} \tag{4}
$$

*Proof.* Note that for $j \geqslant i$ we have:

$$\left\|\frac{\partial \mathbf{v}_j^{(\ell+1)}}{\partial \mathbf{v}_i^{(\ell)}}\right\| = \left\|\frac{\partial}{\partial \mathbf{v}_j^{(\ell)}}\left[\boldsymbol{\psi}^{(\ell)}\left(\text{norm}_2^{(\ell)}\left(\mathbf{z}_j^{(\ell)}\right)\right) + \mathbf{z}_j^{(\ell)}\right]\right\|$$

$$\leqslant \left(\frac{\sigma_{\psi^{(\ell)}}}{\beta_2^{(\ell)}} + 1\right)\frac{\partial \mathbf{z}_j^{(\ell)}}{\partial \mathbf{v}_i^{(\ell)}}$$

$$= \left(\frac{\sigma_{\psi^{(\ell)}}}{\beta_2^{(\ell)}} + 1\right)\frac{\partial}{\partial \mathbf{v}_i^{(\ell)}}\left[\sum_{j\leqslant i}\alpha_{ij}^{(\ell)}\,\text{norm}_1^{(\ell)}\left(\mathbf{v}_i^{(\ell)}\right) + \mathbf{v}_i^{(\ell)}\right]$$

$$= \left(\frac{\sigma_{\psi^{(\ell)}}}{\beta_2^{(\ell)}} + 1\right)\left(\frac{\alpha_{j,i}^{(\ell)}}{\beta_1^{(\ell)}} + \delta_{j,i}\right)$$

where we let $\beta_i^{(\ell)}$ represent the effect of layer normalization $i$ at the $\ell$-th layer and $\sigma_{\psi^{(\ell)}}$ the Lipschitz constant of $\boldsymbol{\psi}^{(\ell)}$. For the case when $j < i$ due to the causal mechanism we have that $\partial \mathbf{v}_j^{(\ell)}/\partial \mathbf{v}_i^{(\ell-1)} = 0$. We compute the following bound:

$$\left\|\frac{\partial \mathbf{y}_n}{\partial \mathbf{v}_i^{(0)}}\right\| = \left\|\frac{1}{\beta_3}\sum_{k_1}\cdots\sum_{k_L}\frac{\partial \mathbf{v}_n^{(L)}}{\partial \mathbf{v}_{k_L}^{(L-1)}}\prod_{\ell=2}^{L-1}\frac{\partial \mathbf{v}_{k_\ell}^{(\ell)}}{\partial \mathbf{v}_{k_{\ell-1}}^{(\ell-1)}}\frac{\partial \mathbf{v}_{k_1}^{(1)}}{\partial \mathbf{v}_i^{(0)}}\right\|$$

$$= \left\|\frac{1}{\beta_3}\sum_{k_1\geqslant i}\cdots\sum_{k_L\geqslant k_{L-1}}\frac{\partial \mathbf{v}_n^{(L)}}{\partial \mathbf{v}_{k_L}^{(L-1)}}\prod_{\ell=2}^{L-1}\frac{\partial \mathbf{v}_{k_\ell}^{(\ell)}}{\partial \mathbf{v}_{k_{\ell-1}}^{(\ell-1)}}\frac{\partial \mathbf{v}_{k_1}^{(1)}}{\partial \mathbf{v}_i^{(0)}}\right\|$$

$$\leqslant \frac{1}{\beta_3}\prod_{\ell=1}^{L}\left(\frac{\sigma_{\psi}}{\beta_2^{(\ell)}} + 1\right)\sum_{k_1\geqslant i}\cdots\sum_{k_L\geqslant k_{L-1}}\bar{\alpha}_{n,k_L}^{(L-1)}\prod_{\ell=2}^{L-1}\bar{\alpha}_{k_\ell,k_{\ell-1}}^{(\ell-1)}\bar{\alpha}_{k_1,i}^{(0)}$$

$$= C\sum_{k_1\geqslant i}\cdots\sum_{k_L\geqslant k_{L-1}}\bar{\alpha}_{n,k_L}^{(L-1)}\prod_{\ell=2}^{L-1}\bar{\alpha}_{k_\ell,k_{\ell-1}}^{(\ell-1)}\bar{\alpha}_{k_1,i}^{(0)}$$

where we let $\bar{\alpha}_{j,i}^{(\ell)} = \frac{\alpha_{j,i}^{(\ell)}}{\beta_1^{(\ell)}} + \delta_{j,i}$ and $C = \frac{1}{\beta_3}\prod_{\ell=1}^{L}\left(\frac{\sigma_{\psi}}{\beta_2^{(\ell)}} + 1\right)$. □

We note that in this derivation, we use simplifying assumptions on the layer norms and attention coefficients, more specifically we assume that they are independent of the $\mathbf{v}_i$s. Of course, there is nothing stopping us from avoiding such assumptions and pushing the partial derivatives inside these components as well. The drawback is that this would add a great deal of additional complexity to the result and potentially distract from what we believe are the two key takeaways: (1) the position of the token matters, and (2) the attention coefficients matter.

**Connection to the spectral theory of Markov chains.** We now show some results on the spectral theory of matrices which relate to causal attention mechanisms. We emphasize that in this work, we view causal attention mechanisms as triangular row-stochastic matrices. We show that these matrices have interesting spectral properties.

**Lemma B.6.** *A row-stochastic triangular matrix* $\mathbf{A}$ *has* 1 *as its largest eigenvalue. Moreover, such eigenvalue has multiplicity* 1 *if each row except the first has at least* 2 *non-zero entries.*

*Proof.* We start by showing that $\mathbf{A}$ cannot have eigenvalues $\lambda > 1$. We then provide an eigenvector with eigenvalue 1. We finally show that such an eigenvector is unique if each row has at least 2 non-zero entries.

Assume $\lambda > 1$ for some eigenvector $\phi$, we then have that $\mathbf{A}\phi = \lambda\phi$. Consider $\phi_i = \max_k \phi_k > 0$. Now $(\mathbf{A}\phi)_i = \sum_{j\leqslant i}\mathbf{A}_{ij}\phi_j = \lambda\phi_i$. As the sum is a convex combination, the result cannot

be larger than the already maximal element $\phi_i$. As $\lambda > 1$, we however have that $\lambda\phi_i > \phi_i$ which is a contradiction and we conclude that $\lambda \leqslant 1$.

It is easy to find an eigenvector that always has eigenvalue 1. Consider a vector $\mathbf{x}$ which is a constant vector of 1s. Then $(\mathbf{Ax})_i = \sum_{j \leqslant i} \mathbf{A}_{ij} = \mathbf{x}_i$, therefore $\mathbf{x}$ is an eigenvector with eigenvalue 1.

Finally, we show that when each row is non-zero, the only eigenvector is the constant-valued eigenvector. Consider the largest entry $\mathbf{y}_i > 0$, then we have that $(\mathbf{Ay})_i = \sum_{j \leqslant i} \mathbf{A}_{ij}\mathbf{y}_j = \mathbf{y}_i$. Again, as this defines a convex combination, we must have that all tokens that $i$ points to (i.e. the non-zero entries) are also equal to $\mathbf{y}_i$. The condition that each row has at least two non-zero entries is important as it means that the condition $\mathbf{y}_i = \mathbf{y}_j$ is true for all tokens. $\square$

**Lemma B.7.** *The product of two row-stochastic matrices is again row-stochastic. Moreover, the product of two triangular row-stochastic matrices is a triangular row-stochastic matrix.*

*Proof.* Let $\mathbf{A}, \mathbf{B} \in \mathbb{R}^{n \times n}$ be two row-stochastic matrices. We compute:

$$\sum_j (\mathbf{AB})_{ij} = \sum_j \sum_k \mathbf{A}_{ik}\mathbf{B}_{kj} = \sum_k \mathbf{A}_{ik} \sum_j \mathbf{B}_{kj} = 1$$

The final statement follows immediately from the fact that the product of two triangular matrices is triangular. $\square$

We now show that under specific conditions, our over-squashing bound converges to a steady state in which the final token $\mathbf{y}_n$ only depends on the initial input token $\mathbf{v}_1^{(0)}$ as the number layers tends to infinity, i.e. $L \to \infty$.

**Proposition B.8.** *Let $\beta_1^{(\ell)}, \beta_2^{(\ell)} = 1$, $\beta_3^{1/L} = 4$, $\sigma_\psi = 1$. Furthermore, for simplicity, let the attention coefficients be equal at each layer and such that each row except the first of the causal mechanism has at least two non-zero elements. Then, we have as $L \to \infty$ that $\partial\mathbf{y}_n/\partial\mathbf{v}_i^{(0)} = 0$ when $i \neq 1$ and $\partial\mathbf{y}_n/\partial\mathbf{v}_i^{(0)} = 1$ when $i = 1$. In other words, $\mathbf{y}_n$ will only be sensitive to the first token.*

*Proof.* Let the associated attention matrix be $\mathbf{\Lambda}$. We start by re-writing the following:

$$\left\| \frac{\partial\mathbf{y}_n}{\partial\mathbf{v}_i^{(0)}} \right\| \leqslant \frac{1}{\beta_3} \prod_{\ell=1}^{L} \left( \frac{\sigma_\psi}{\beta_2^{(\ell)}} + 1 \right) \sum_{k_1 \geqslant i} \cdots \sum_{k_L \geqslant k_{L-1}} \bar\alpha_{n,k_L}^{(L-1)} \prod_{\ell=2}^{L-1} \bar\alpha_{k_\ell,k_{\ell-1}}^{(\ell-1)} \bar\alpha_{k_1,i}^{(0)}$$

$$= \left( \prod_{\ell=1}^{L} \left[ \frac{1}{\beta_3^{1/L}} \left( \frac{\sigma_\psi}{\beta_2} + 1 \right) \left( \frac{1}{\beta_1}\mathbf{\Lambda} + \mathbf{I} \right) \right] \right)_{n,i}$$

$$= \left( \left[ \frac{1}{2}(\mathbf{\Lambda} + \mathbf{I}) \right]^L \right)_{n,i}$$

We now point out that $\tilde{\mathbf{\Lambda}} = \frac{1}{2}(\mathbf{\Lambda} + \mathbf{I})$ is row-stochastic and with our assumptions is diagonalizable into $\tilde{\mathbf{\Lambda}} = \Psi\Sigma\Phi$. In particular, by Lemma B.7, also $\tilde{\mathbf{\Lambda}}^L$ is row-stochastic and each entry is non-negative. We now use the Perron-Frobenius theorem for non-negative matrices [23], which guarantees us that all eigenvalues $\lambda_k$ of $\tilde{\mathbf{\Lambda}}$ are bounded such that $|\lambda_k| \leqslant 1$. In particular, thanks to Lemma B.6, we know that there is a unique eigenvector (the constant eigenvector $\psi_n$) with eigenvalue $\lambda_n = 1$. Denote the left eigenvectors by $\psi_k$ and the right eigenvectors $\phi_n$, we therefore have:

$$\lim_{L \to \infty} \tilde{\mathbf{\Lambda}}^L = \sum_k \lambda_k^L \psi_k \phi_k^T = \psi_n \phi_n^T.$$

In particular, one can check that $\phi_n^T = [1 \ 0 \dots \ 0]$, meaning that $\psi_n \phi_n^T$ has as first column a constant vector of 1s and every other entry 0. This completes the proof. $\square$

### B.3 Counting

We finally show in this section our final results that apply specifically to counting tasks. We start by highlighting a potential difficulty that the softmax layer encounters when counting, namely that the normalisation used makes it hard for it to preserve a notion of magnitude present in the sequence.

**Proposition B.9.** *A Transformer without positional encodings and a causal attention mechanism is immediately unable to solve the counting problem.*

*Proof.* We show this statement by demonstrating that the only information preserved about the 'count' by the attention mechanism will be the ratio of the elements present in a sequence. In particular, sequences with the same ratio of tokens will be assigned the exact same representation — this applies as we specifically study an attention mechanism without positional encodings and causal masking. Of course, having the same ratio of elements does not mean that the count will be the same, for instance the sequences '10' and '1100' have the same ratio of digits but clearly different counts.

Consider a sequence of two values, $\mathbf{v}_{zero}^{(0)}$ and $\mathbf{v}_{one}^{(0)}$, with $n_0$ and $n_1$ being the number of zeros and ones respectively. We ignore in our calculations the MLPs $\psi$ and the normalizations norm as these don't affect the argument. As this specific attention mechanism is permutation equivariant, the initial zero tokens will all be mapped to:

$$
\begin{aligned}
\mathbf{z}_{zero}^{(1)} &= \sum_j \frac{\exp\left(\mathbf{q}_{zero}^{(0)T}\mathbf{k}_j^{(0)}\right)}{\sum_w \exp\left(\mathbf{q}_{zero}^{(0)T}\mathbf{k}_w^{(0)}\right)}\mathbf{v}_j^{(0)} + \mathbf{v}_{zero}^{(0)} \\[2mm]
&= \frac{n_0\exp\left(\mathbf{q}_{zero}^{(0)T}\mathbf{k}_{zero}^{(0)}\right)}{n_0\exp\left(\mathbf{q}_{zero}^{(0)T}\mathbf{k}_{zero}^{(0)}\right) + n_1\exp\left(\mathbf{q}_{zero}^{(0)T}\mathbf{k}_{one}^{(0)}\right)}\mathbf{v}_{zero}^{(0)} + \frac{n_1\exp\left(\mathbf{q}_{zero}^{(0)T}\mathbf{k}_{one}^{(0)}\right)}{n_0\exp\left(\mathbf{q}_{zero}^{(0)T}\mathbf{k}_{zero}^{(0)}\right) + n_1\exp\left(\mathbf{q}_{zero}^{(0)T}\mathbf{k}_{one}^{(0)}\right)}\mathbf{v}_{one}^{(0)} + \mathbf{v}_{zero}^{(0)} \\[2mm]
&= \frac{\exp\left(\mathbf{q}_{zero}^{(0)T}\mathbf{k}_{zero}^{(0)}\right)}{\exp\left(\mathbf{q}_{zero}^{(0)T}\mathbf{k}_{zero}^{(0)}\right) + \frac{n_1}{n_0}\exp\left(\mathbf{q}_{zero}^{(0)T}\mathbf{k}_{one}^{(0)}\right)}\mathbf{v}_{zero}^{(0)} + \frac{\exp\left(\mathbf{q}_{zero}^{(0)T}\mathbf{k}_{one}^{(0)}\right)}{\frac{n_0}{n_1}\exp\left(\mathbf{q}_{zero}^{(0)T}\mathbf{k}_{zero}^{(0)}\right) + \exp\left(\mathbf{q}_{zero}^{(0)T}\mathbf{k}_{one}^{(0)}\right)}\mathbf{v}_{one}^{(0)} + \mathbf{v}_{zero}^{(0)}
\end{aligned}
$$

Similarly, the ones will be mapped to:

$$
\begin{aligned}
\mathbf{z}_{one}^{(1)} &= \sum_j \frac{\exp\left(\mathbf{q}_{one}^{(0)T}\mathbf{k}_j^{(0)}\right)}{\sum_w \exp\left(\mathbf{q}_{one}^{(0)T}\mathbf{k}_w^{(0)}\right)}\mathbf{v}_j^{(0)} + \mathbf{v}_{one}^{(0)} \\[2mm]
&= \frac{n_0\exp\left(\mathbf{q}_{one}^{(0)T}\mathbf{k}_{zero}^{(0)}\right)}{n_0\exp\left(\mathbf{q}_{one}^{(0)T}\mathbf{k}_{zero}^{(0)}\right) + n_1\exp\left(\mathbf{q}_{one}^{(0)T}\mathbf{k}_{one}^{(0)}\right)}\mathbf{v}_{zero}^{(0)} + \frac{n_1\exp\left(\mathbf{q}_{one}^{(0)T}\mathbf{k}_{one}^{(0)}\right)}{n_0\exp\left(\mathbf{q}_{one}^{(0)T}\mathbf{k}_{zero}^{(0)}\right) + n_1\exp\left(\mathbf{q}_{one}^{(0)T}\mathbf{k}_{one}^{(0)}\right)}\mathbf{v}_{one}^{(0)} + \mathbf{v}_{one}^{(0)} \\[2mm]
&= \frac{\exp\left(\mathbf{q}_{one}^{(0)T}\mathbf{k}_{zero}^{(0)}\right)}{\exp\left(\mathbf{q}_{one}^{(0)T}\mathbf{k}_{zero}^{(0)}\right) + \frac{n_1}{n_0}\exp\left(\mathbf{q}_{one}^{(0)T}\mathbf{k}_{one}^{(0)}\right)}\mathbf{v}_{zero}^{(0)} + \frac{\exp\left(\mathbf{q}_{one}^{(0)T}\mathbf{k}_{one}^{(0)}\right)}{\frac{n_0}{n_1}\exp\left(\mathbf{q}_{zero}^{(0)T}\mathbf{k}_{zero}^{(0)}\right) + \exp\left(\mathbf{q}_{one}^{(0)T}\mathbf{k}_{one}^{(0)}\right)}\mathbf{v}_{one}^{(0)} + \mathbf{v}_{one}^{(0)}
\end{aligned}
$$

Assuming that $\mathbf{z}_{zero}^{(1)} \neq \mathbf{z}_{one}^{(1)}$ (to avoid the trivial case), we notice that the attention mechanism alongside the MLP $\psi$ define an isomorphism between sequences at different layers, updating all zeros and ones to a different value vector. The critical fact is that the representations *only depend on the ratio between $n_0$ and $n_1$*, meaning that sequences of different lengths (therefore different counts) will have the exact same representation. This is respected at each layer, meaning that the LLM after $L$ layers will assign the same representation to different

sequences as long as they have the same ratio. This points to a loss of representation for the counting problem. □

**Corollary B.10.** *Consider a task in which the goal is to count how many* $\mathbf{v}_a$ *tokens there are in the sequence. Let* $\mathbf{v}^{(0)} = \begin{bmatrix} \mathbf{v} & \mathbf{v}_a \end{bmatrix}^T \in \mathbb{R}^{n \times d}$ *and* $\mathbf{v}^{*(0)} = \begin{bmatrix} \mathbf{v} & \mathbf{v}_a & \mathbf{v}_a \end{bmatrix} \in \mathbb{R}^{(n+1) \times d}$. *Due to representational collapse, at least one sequence will be given the wrong count for large enough finite* $n$.

*Proof.* This statement is a direct consequence of representational collapse. In particular, as $\mathbf{y}_n$ and $\mathbf{y}_n^*$ will be indistinguishable for large enough $n$, the Transformer will be forced to make a mistake for at least one of them. This points to an impossibility result of counting on certain sequences due to floating point error. This holds regardless of the positional encodings used (as long as they satisfy the required decay conditions) and causal mechanism. □

# C  Experiments

The prompting done on Gemini 1.5 in our work does not require custom resources as we use hosted Gemini instances. We run a local version of Gemma 7B on modest hardware to analyse the internal representations.

## C.1  Experimental Details

We detail the way in which we execute the prompting for the various experiments.

**Counting experiments.**  For the sum experiment we prompt as:
```
Please perform the following sum:  seq.  Please give the answer on the
final line exactly as 'The final answer to your maths question is:  xxxx',
where 'xxxx' is your answer..
```
For the ones and zero sequences, we similarly prompt as
```
Please count the number of ones in the following sequence:seq.  Please give
the answer on the final line exactly as 'The final answer to your maths
question is:  xxxx', where 'xxxx is your answer.
```
For the word counting experiment, we prompt as
```
Please count the number of times 'car' appears in the following sentence:
'seq'.  Please give the answer on the final line exactly as 'The final
answer to your maths question is:  xxxx', where 'xxxx' is your answer.
```
For the CoT experiments, we supply examples of the form:

```
Let's think step by step, showing me your reasoning.  Here are a few
examples:
Please perform the following sum:  1 + 1 + 1 + 1 + 1 + 1 + 1 + 1 + 1 + 1 +
1 + 1
We divide the sum into groups of 5.  (1 + 1 + 1 + 1 + 1) + (1 + 1 + 1 + 1 +
1) + 1 + 1
The answer is then 2 * 5 + 2 = 12
The final answer to your maths question is:  12
Please perform the following sum:  1 + 1 + 1 + 1 + 1 + 1
We divide the sum into groups of 5.
(1 + 1 + 1 + 1 + 1) + 1
The answer is then 1 * 5 + 1 = 6
The final answer to your maths question is:  6
```
With similar strategies for the 4 experiments.

**Copying experiments.**  For the copying experiments, we use the following prompt:
```
Consider the following sequence:  seq.  What is the last digit in this
sequence?  Please answer exactly as 'The answer to your question is:
<ANSWER>'
```
and change appropriately the sequence as described.

We commit to releasing the code we have used to generate the prompts in the near future.

## C.2 Counting with Gemma

We report similar results for the counting experiments using Gemma [27] in Figure 6 and 7. Compared to Gemini 1.5, Gemma seems to answer less accurately on the counting prompts.

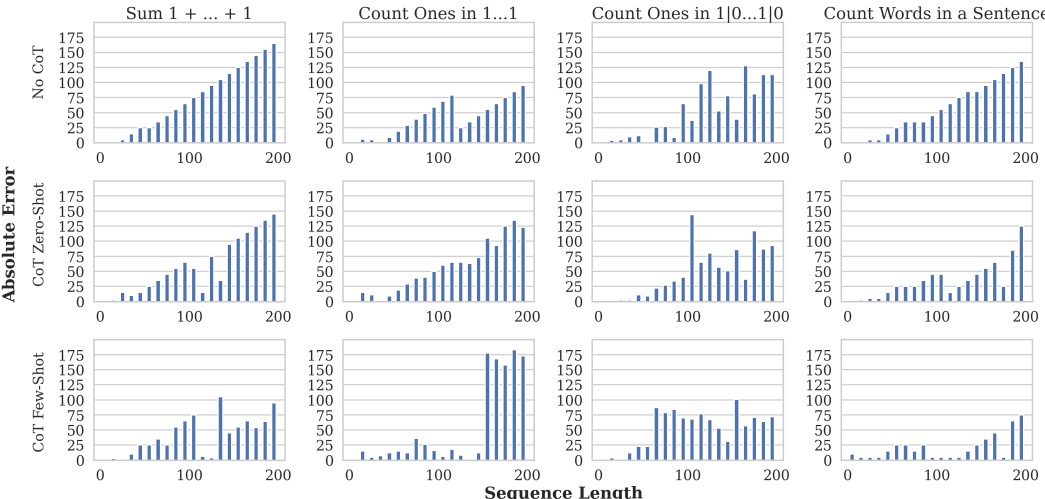

Figure 6: Gemma 7B LLMs being prompted to (i) sum $1 + \cdots + 1$ (left), (ii) Count the number of ones in a sequence of 1s (center), and (iii) Count the number of ones in a sequence of ones and zeroes (the sequence is a Bernoulli sequence with probability of sampling a one being 0.7) (right).

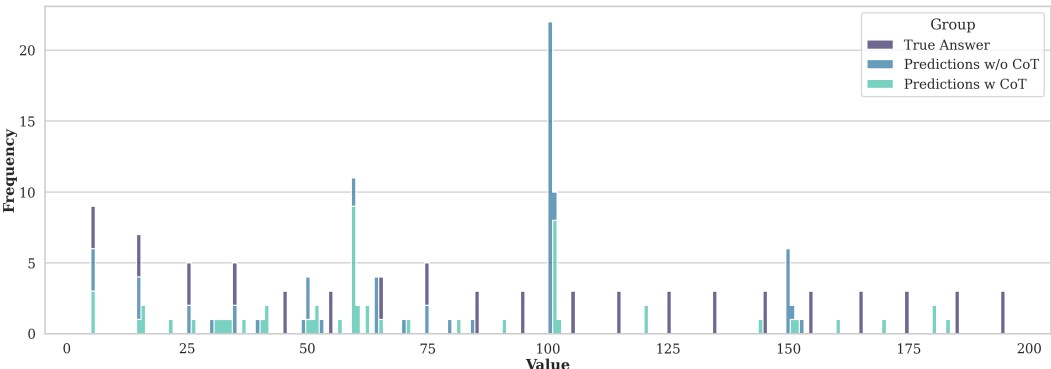

Figure 7: Frequency of different outputs for Gemma 7B

## C.3 Synthetic Experiments on Representational Collapse

To provide further experimental evidence of representational collapse with other positional encodings, we experiment using the original sinusoidal embeddings from [30]. We sample key, query, and values from a Gaussian distribution with variance $\sigma^2 = 1/d$, with $d$ the dimension of the embeddings. We set $d = 64$ and otherwise follow the exact structure of the decoder-only Transformer presented in the original Transformer paper. We experiment with a single attention layer and check the convergence of the representations of the final token between a sequence of length $n$ and a sequence of length $n + 1$ in which we simply copy the final token. We present the results in Figure 8. We see how also for sinusoidal PEs with key, queries, and values randomly sampled, the convergence still occurs.

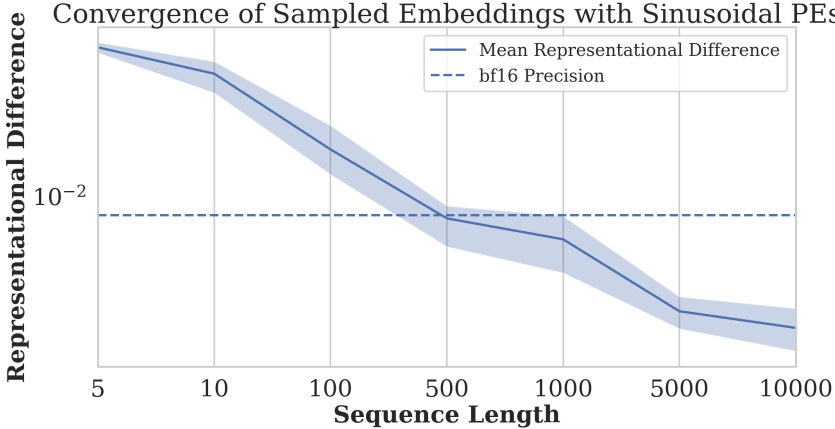

Figure 8: Convergence behaviour with a synthetic Transformer experiment. We sample the key, query, and values from a Gaussian distribution and apply the traditional sinusoidal PEs from [30]. We apply a logarithmic scale on the y-axis.

Finally, we test the decay of the total variation of the softmax distributions of two growing sequences, to experimentally verify Lemma B.2. We sample a sequence $\mathbf{x}$ of length $n$ with values uniformly distributed in the range $[0, 1]$. We then create $\mathbf{x}^*$ by adding to the first $k = 200$ elements of $\mathbf{x}$ noise which is uniformly sampled between $[0, 0.1]$. In Figure 9, we show how the total variation between their respective softmax distributions decays with the sequence length.

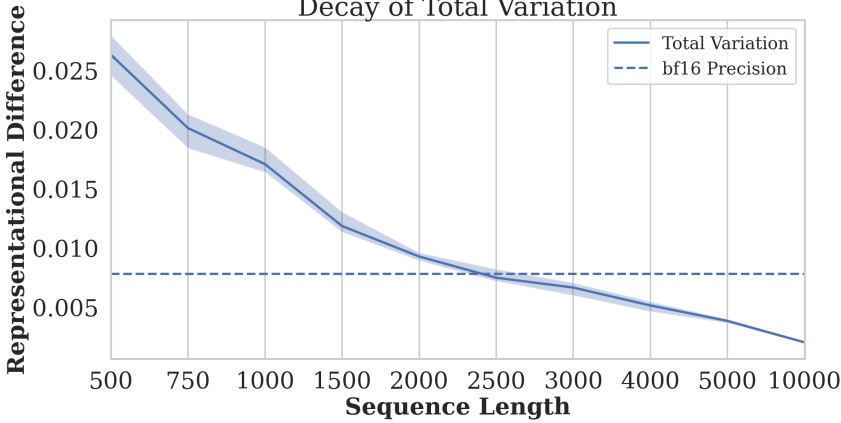

Figure 9: Total variation decay of softmax distributions with growing sequence length. We sample $n$ elements uniformly from $[0, 1]$ and then create a related sequence by taking its first $k = 200$ and adding to these elements noise sampled uniformly from $[0, 0.1]$. We measure the total variation between their softmax distributions. It is clear how the total variation decays with length, in accordance with Lemma B.2. Error bars show minimum and maximum over 5 seeds.

## C.4 Effect of Positional Encodings

We include a synthetic experiment where we verify the occurrence of the representational collapse phenomenon with different Positional Encodings, namely: Alibi [24], the original Absolute Positional Encodings (APE) [30], and No Positional Encodings (NoPE) [15]. In our synthetic experiment, we sample $n$ queries and keys independently directly from a standard Gaussian. We then construct a related sequence of length $n + 1$ by repeating its last element. We report the $L_1$ distance between the two sequences after a single decoder-Transformer layer, as done for the other representational collapse experiments. We consider a Transformer with

a hidden dimension of 64, a single attention head, and we apply normalisations to simulate layer norm. The Transformer is not trained, but only used to simulate the propagation of information of queries and keys sampled from a Gaussian distribution. The results are shown in Figure 10. Representational collapse seems to occur with all 4 positional encodings, with the convergence of the representations happening at a similar sequence lengths.

We would like to highlight that our condition on the decay of RoPE necessary to fulfill the requirement of Theorem 4.2 is inspired by claims of the decay of RoPE coming from the original work by Su et al. [26]. However, recent work has shown that such claims may not be strictly always upheld [4], as the original claims relied on very specific conditions on the queries and keys. We are of the opinion; however, that the range of synthetic and real-world experiments in this work support our representational collapse claims in practice with RoPE. A more precise mathematical treatment of RoPE specifically is therefore left as future work.

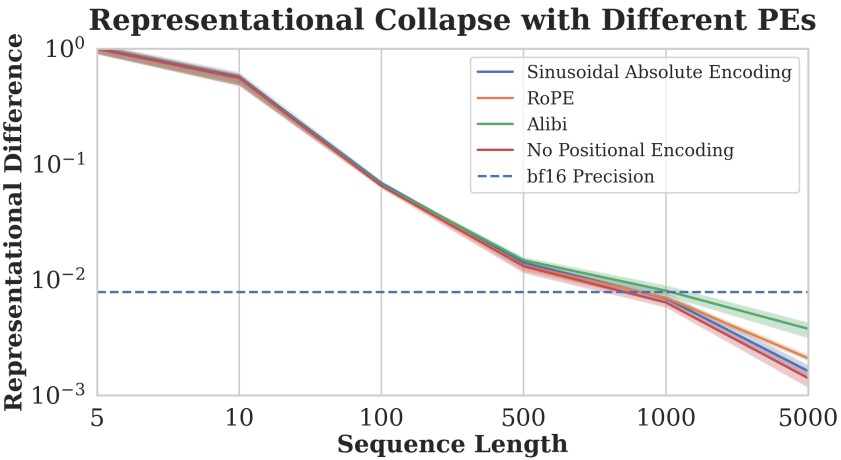

Figure 10: We sample $n$ queries, keys, and values independently from a standard Gaussian, applying different positional encodings. We then construct sequences of length $n + 1$, by repeating the $n$-th token. We report the $L_1$ distance between the last tokens of the sequences of length $n$ and $n + 1$ after one decoder-only Transformer layer. We set the hidden dimension to 64, use a single attention head, and normalise appropriately to simulate the effects of LayerNorm. The y-axis is shown in log-scale.

### C.5 Ablation on Prompt Structure

We ablate the prompt structure specifically for the copying task. In particular, we consider the prompts: (Type 1) "What is the last digit of the following sequence? Please answer exactly as 'The answer to your question is: <ANSWER>' ". Here is the sequence: {seq} and (Type 2) "Please answer exactly as 'The answer to your question is: <ANSWER>'. What is the last digit of the following sequence? {seq}". The results are presented in Figure 11. We find that the prompt indeed does affect the performance on the task, as the prompt affects the distribution of the attention over the layer. However, for both types of prompts, the model ends up failing, in accordance with our theory. We also show for completeness, in Figure 12, that representational collapse occurs in Gemma 7B also for the 'Type 1' prompt.

### C.6 Local sliding window attention

A fundamental limitation of an attention mechanism that leverages the softmax function is that it cannot remain sharp, especially as the sequence length grows [31]. This is in fact a key intuition that we exploit to show our result on representational collapse. A good way to address representational collapse and the related phenomenon of over-squashing is then that of limiting the spread of the softmax function, by directly limiting the amount of tokens the attention mechanism pays attention to. This mechanism is often referred to as a *local sliding window* and is a major architectural change present in Gemma 2 [28]. We believe that such an architectural change elegantly addresses representational collapse and over-squashing at

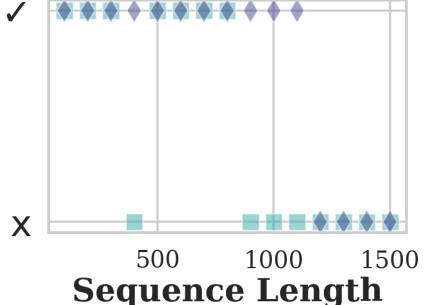

Figure 11: Performance of a Gemini model on the following prompts: (Type 1) "What is the last digit of the following sequence? Please answer exactly as 'The answer to your question is: <ANSWER>' ". Here is the sequence: {seq} and (Type 2) "Please answer exactly as 'The answer to your question is: <ANSWER>'. What is the last digit of the following sequence? {seq}"

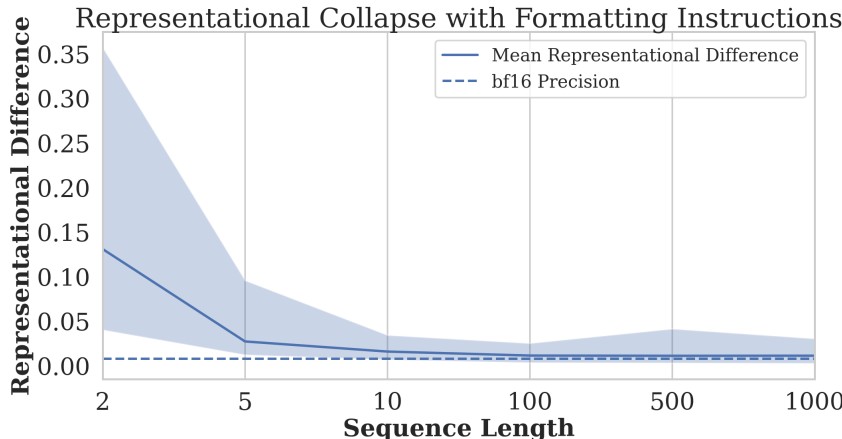

Figure 12: Representational collapse in Gemma for the prompt: "What is the last digit of the following sequence? Please answer exactly as 'The answer to your question is: <ANSWER>' ". Here is the sequence: {seq} and (Type 2) "

the source as it avoids the issues that come with growing token sequences – something which our theory often exploits.

