# OpenReview forum: "Transformers need glasses! Information over-squashing in language tasks"
_NeurIPS.cc/2024/Conference — NeurIPS 2024 poster_

### Official Review · Reviewer_vhDf · 2024-07-06

**Soundness:** 3
**Presentation:** 4
**Contribution:** 3
**Rating:** 7
**Confidence:** 3

**Summary:**

This paper presents an in-depth analysis of decoder-only Transformers, focusing on their limitations in handling information propagation. The authors identify two key phenomena: "representational collapse" due to "over-squashing" (line in Graph Neural Networks). These issues lead to a significant loss of information, especially in tasks involving counting and copying, which require precise handling of individual tokens. The paper combines theoretical analysis with empirical evidence to demonstrate these problems.

**Strengths:**

1. The paper provides a theoretical framework to understand the limitations of decoder-only Transformers. The concepts of representational collapse and over-squashing are well-formulated and offer valuable insights into why Transformers struggle with certain tasks.
2. Very strong theoretical analysis is provided in paper, but the clarity of narrative is preserved. The quality of presentation is excellent.
3. The authors provide empirical evidence from contemporary LLM, specifically Gemini 1.5 and Gemma 7B, supporting their theory. They also provide the analysis of effect of floating point precision.
4. Authors carefully provide all the necessary details about experiment.

**Weaknesses:**

1. The experiments are primarily focused on specific artificial tasks (counting and copying), but the study lacks the analysis of the real-world texts. It would be beneficial to include some statistics of difference of tokens' representations on the standard text corpora.
2. Authors provide only one artificial solution to split the sequence by different token.
3. The theoretical analysis makes some simplifying assumptions, such as treating attention weights as independent of the input. While the authors justify these assumptions, it would be beneficial to explore their impact on the results more thoroughly.

**Questions:**

1. Could you please include some statistics of difference of tokens' representations on the standard text corpora? It would be interesting to see does representational collapse happen in natural texts.
2. How can you deal with representational collapse in real-world scenarios? Wouldn't it be worse to include random token for the information?
3. Could this theoretical result be generalized to non-causal language modeling (different attention mask)?

Also here are some small remarks:

Line 331 typo: summariesd

Line 339 space in "phenomena : representational"

**Limitations:**

1. Theoretical analysis relies on several simplifying assumptions, which are justified but may not fully capture the complexities of real-world models.
2. Empirical validation is conducted only on two specific models

---

> ### Author Rebuttal · Authors · 2024-08-06
>
> We are happy to hear that you believe our paper offers valuable insights into Transformers, has a very strong theoretical analysis, and excellent presentation. We would like to address your questions.
>
> **(Q1)** *Could you please include some statistics of difference of tokens' representations on the standard text corpora? It would be interesting to see does representational collapse happen in natural texts.*
>
> We thank you for the thought-provoking question!
>
> Having given it consideration, we believe that it is not sufficiently well-defined to measure token representation difference on generic natural text prompt pairs.
>
> Generally, representational collapse can only be rigorously talked about w.r.t. a specific family of prompts which are all related by a certain property. Throughout our paper we explore a number of these.
>
> Hence, to rigorously study representational collapse in general-purpose corpora, we would first need to have a method to detect specific subsets of inputs in these corpora that form a relevant family, and we find this to be a non-trivial endeavor beyond the scope of this rebuttal. Randomly-chosen pairs of natural prompts without clear relations to each other are highly unlikely to have collapsing representations due to the high diversity of tokens in both.
>
> All this being said, we believe there are many domains where representational collapse, even just over families of prompts, can be a significant issue---some key examples include: (1) finance where it is common to find numbers with many zeros (share counts, acquisition amounts, etc…), (2) numerical spreadsheets, (3) LLMs operating over bit sequences. This list is of course not exhaustive, but provides some concrete examples in which our analysis would be likely very relevant.
>
> **(Q2)** *How can you deal with representational collapse in real-world scenarios? Wouldn't it be worse to include random token for the information?*
>
> In general, we believe that it is very hard to avoid the issue of representational collapse without modifying the architecture, or as you hinted in the next question, the attention mask. Solutions such as adding commas between repeated digits could work well for domains such as finance, where this is common practice to help parse large numbers. Similarly, adding “whitespace” or even avoiding repetition due to tokenization (as we touch upon in our paper)  will have the same effect. We believe that adding random tokens is likely to do more harm than good, but adding “filler” tokens is instead likely a practical solution, something that has already been observed to help in certain cases [2].
>
> **(Q3)** *Could this theoretical result be generalized to non-causal language modeling (different attention mask)?*
>
> This is an excellent point and something that we think would be very important to comment on in our paper. Indeed different attention masks (such as the commonly used sliding-window attention), are going to behave differently to a full causal mask, both in regards to representational collapse and over-squashing. A sliding-window mask for instance can help with representational collapse as now the contribution of a single token is not forced to monotonically decrease with sequence length. It would also help fix the topological aspect of over-squashing in which tokens at the beginning of the sequence are favored, in fact it would flip this. We will add a more detailed analysis of this in a new section in the appendix, touching upon sliding-window attention and the common alternation between windowed (local attention) and standard causal attention as done in the new Gemma 2 models [1], for instance.  We once again thank you very much for this great comment and are happy to provide more details of our analysis.
>
> We hope to have addressed your questions and point you to the global comment for additional ablations and discussions. We thank you once again for endorsing our paper. We are looking forward to the rebuttal period.
>
> [1] Gemma 2: Improving Open Language Models at a Practical Size. Google DeepMind, ArXiv, 2024.
>
> [2] Let’s Think Dot by Dot: Hidden Computation in Transformer Language Models. Pfau et al, ArXiv, 2024.

---

> > ### Comment · Reviewer_vhDf · 2024-08-12
> >
> > Thank you for your quick and very detailed reply. My concerns were answered, I understand now the real-world scenarios, and I admire the filler-token solution supported by references. I believe the score is already high enough, but you have fully answered all the questions. I am looking forward to see the detailed analysis of several types of attention in appendix.
> >
> > Best wishes to your paper, and let me know if I can be of any help.

---

> > > ### Author Response · Authors · 2024-08-12
> > > **Thank you!**
> > >
> > > Thank you for acknowledging our responses and appreciating our work! We are delighted to hear our comments answered your concerns. Should you have any other questions, please feel free to let us know.

---

### Official Review · Reviewer_GKd9 · 2024-07-09

**Soundness:** 3
**Presentation:** 3
**Contribution:** 3
**Rating:** 6
**Confidence:** 3

**Summary:**

The paper provides a theoretical and empirical analysis of the final representations of transformers, revealing the phenomenon of representational collapse.

**Strengths:**

- The paper is well-written.
- It highlights a problem in transformers that causes them to fail on a large set of tasks (assuming this extends to addition, etc.).
- The paper provides empirical evidence supporting the theoretical claims, demonstrating the real-world relevance of the identified issues.
- The impact of low-precision floating-point formats is interesting, which is highly relevant to current everyday practices

**Weaknesses:**

- There is no strong solution to this problem. Although the authors provide a simple approach, it is challenging to understand how to apply this approach in a practical setting.
- The problems seem to be tied to the positional embeddings, as the authors stated. NoPE (https://arxiv.org/abs/2305.19466) and no causal mask would make this problem impossible. However, there are no experiments on how certain positional embeddings might be better than others.

**Questions:**

- What do the authors believe is the breadth of tasks affected by this fundamental issue in transformers?
- What are the positional embeddings used for the experimental settings? Can these affect the representational collapse?
- Minor: Some figures, like Figure 2, have a very small font size, making them hard to read.

**Limitations:**

See Questions and Weaknesses.

---

> ### Author Rebuttal · Authors · 2024-08-06
>
> We thank you for pointing out that our paper is well written and that you found it interesting and with real-world relevance. We would like to address your questions on the breadth of tasks affected and on positional encodings.
>
> **(Q1)** *What do the authors believe is the breadth of tasks affected by this fundamental issue in transformers?*
>
> While it is hard to comment on the entirety of the tasks affected by this issue, we believe that copying and counting are fundamental enough to claim that this is a rather wide-spread issue concerning the repetition of tokens. Copying is especially important as many LLM endpoints today benefit from tool-use, which relies on copying.
>
> We suspect that tasks that see frequent repetition of tokens to be particularly problematic. For instance in finance it is common to see many consecutive 0s. For such a case we believe that our simple solution of including “,” every third digit is rather sensible and practical. We kindly refer to the global comment for a longer discussion on what we believe are the applications of our results.
>
> **(Q2)** *What are the positional embeddings used for the experimental settings? Can these affect the representational collapse?*
>
> We thank you for the valuable question. Our experiments rely on the positional encodings used by Gemma (RoPE). We believe that RoPE is a wide-spread PE, for instance also used by Llama 3 [1]. Having said this, there is no particular reason that this issue is one of RoPE specifically. To demonstrate this, in the global comment we describe an ablation we have done showcasing representational collapse occurring with RoPE, Alibi [2], original sinusoidal embeddings (APE) [3], and No positional encodings (NoPE) [4].  We refer to the global comment for further details and to the supplementary PDF for the results.
>
> **(Q3)** *Minor: Some figures, like Figure 2, have a very small font size, making them hard to read.*
>
> We thank you for pointing this out, we have now improved the readability and font size of the figures throughout the paper.
>
> We hope that our response and additional experiments answer your questions. We are of course more than happy to keep engaging with you and thank you again for endorsing our paper.
>
> [1] The Llama 3 Herd of Models, July 23, 2024. https://ai.meta.com/research/publications/the-llama-3-herd-of-models/.
>
> [2] Train Short, Test Long: Attention with Linear Biases Enables Input Length Extrapolation, Press et al. Arxiv, 2021.
>
> [3] Attention is all you need, Vaswani et al. Neurips, 2017.
>
> [4] The Impact of Positional Encoding on Length Generalization in Transformers, Kazemnejad et al, Neurips, 2023.

---

> ### Comment · Reviewer_GKd9 · 2024-08-09
>
> I thank the authors for their reply.
>
> After reading their comments, I will keep my score.  I do believe that this paper helps us understand a problem in LLM that is directly seen in a couple of problems such as copying and counting that has an impact in many applications. However, I did not increase my score further because I do not believe that a strong solution was proposed to this problem.

---

> > ### Author Response · Authors · 2024-08-09
> >
> > We would like to thank you once again for your time and consideration. We are pleased that you wish to maintain your acceptance score for our paper. We are very much excited about the potential of our work to contribute to stronger and scalable solutions in the future.

---

### Official Review · Reviewer_SAa1 · 2024-07-09

**Soundness:** 3
**Presentation:** 4
**Contribution:** 2
**Rating:** 6
**Confidence:** 3

**Summary:**

In the paper, the authors first discuss a phenomenon occurring in LLMs that they call "representational collapse". They provide empirical evidence of the phenomenon in state-of-the-art language models and they provide a theoretical justification for it. They then show that decoder-only transformers exhibit what is known in graph network theory as "over-squashing", which may cause loss of information in the final prediction.

**Strengths:**

The paper highlights a curious phenomenon in current language models, which shows their limitations even for extremely simple tasks like counting. The paper is well organized and well written. The observation of the over-squashing phenomenon is interesting.

**Weaknesses:**

The observed phenomenon, while interesting, has limited scope. Also, the representational collapse result is hardly surprising to me. It feels quite trivial that in the limit of a very long sequence, the importance of a single token out of many repeated ones becomes negligible.

**Questions:**

1) While the paper claims that the transformer architecture that they analyze is the most popular in state-of-the-art LLMs, it seems to me that they only consider relative positional embeddings and not absolute ones. To the best of my knowledge, current LLMs like GPT and Llama all use absolute positional embeddings. Would your results also extend to this case? In any case, I would highlight the use of relative positional embeddings more prominently in the main text.
2) In the actual prompts used for the experiments, the sequence does not appear at the end of the prompt but somewhere in the middle. Would your theoretical results somehow generalize if the repeated token is not the last one? Would anything change in the experiments if the sequence appears exactly at the end?

---

> ### Author Rebuttal · Authors · 2024-08-06
>
> We are happy that you found our paper well organized, well written, and the over-squashing phenomenon interesting. We would like to address your questions on positional encodings and on the prompting.
>
> **(Q1)** *[...] it seems to me that they only consider relative positional embeddings and not absolute ones. To the best of my knowledge, current LLMs like GPT and Llama all use absolute positional embeddings. Would your results also extend to this case?*
>
> We thank you for the excellent comment. We would like to point out that according to Table 3 from the most recent Llama 3 [1] report, Llama 3 seems to be using the relative positional encoding RoPE. As far as we are aware, details regarding positional encodings used in most recent GPT models are not publicly available and as such we cannot comment with certainty on the positional encodings used for such models.
>
> Regardless, we agree with you that it is important to add more details on the type of positional encoding used. We clarify that our results still apply for absolute positional encodings as we never make any explicit assumptions on the formulation of the PE. What we do require is that the effect of the positional encoding decays with distance, a common design choice used to encode distance in PEs. We have further emphasized this in the main text and will add a detailed section in the Appendix covering RoPE, Alibi [2], absolute sinusoidal embeddings (APE) [3], and no positional encodings (NoPE) [4].
>
> To provide experimental evidence supporting such claims, we have designed a synthetic experiment showing that representational collapse occurs also for Alibi, sinusoidal absolute positional encodings, and NoPE. We refer to the global comment for the details and results. We thank you for helping to strengthen our work in this direction, which we believe is a very important one.
>
> **(Q2)** *In the actual prompts used for the experiments, the sequence does not appear at the end of the prompt but somewhere in the middle. Would your theoretical results somehow generalize if the repeated token is not the last one? Would anything change in the experiments if the sequence appears exactly at the end?*
>
> We thank you for the great point you raise. In our prompts we add the formatting instructions at the end of the prompt in order to help with the automatic parsing of the output. We did not notice any qualitative difference in the counting tasks. On the copying tasks it seems like the formatting affects more the results. To explore this, we reformatted in 2 ways:
>
> 1) "What is the last digit of the following sequence? Please answer exactly as 'The answer to your question is: <ANSWER>'. Here is the sequence: {seq}"
>
> 2) "Please answer exactly as 'The answer to your question is: <ANSWER>'. What is the last digit of the following sequence? {seq}"
>
> The results are in the supplementary PDF in Figure 2, with the first being called Prompt Type 1 and the second Prompt Type 2. We see that the model encounters the same failure for both prompts, with some prompts failing at a later point than others.
>
> The theory applies most directly when the token difference occurs at the end, but it can also be applied to the prompts with the formatting instructions coming at the end. This holds due to a recursive argument. Once the representation has collapsed, the information for that token will be lost in the subsequent tokens as well. To confirm this, we plot the hidden representations of Gemma under the same prompt 3) used in the original copying experiments and show that the representations collapse in such a case – see Figure 3 in the supplementary PDF. We will add these two additional experiments to the appendix of our work.
>
> 3) “Consider the following sequence: {seq} What is the last digit in this sequence? Please answer exactly as 'The answer to your question is: <ANSWER>'”
>
>
> We thank you again for your comment and believe that these additional results help to strengthen our paper in this direction.
>
> **(W1)** *The observed phenomenon, while interesting, has limited scope. Also, the representational collapse result is hardly surprising to me [...]*
>
> While we are happy that you find the phenomenon to be interesting, we respectfully disagree that its scope is limited, as it highlights a fundamental issue of the Transformer architecture to perform important tasks.
>
> We believe that there are two fundamental observations in our analysis of representational collapse that are valuable. (1) We believe that the idea of analysing what can be contained in the embeddings of the last token of the last layer is a novel approach that may provide an interesting new way of studying decoder-only Transformers. (2) We believe that tying representational collapse to an inability to generalize/solve problems like counting and copying due to floating point arithmetic errors is novel. We believe that connecting (1), (2), and representational collapse together creates an interesting way of understanding decoder-only Transformers. We emphasize that tasks such as copying are fundamental for settings in which an AI agent dispatches part of a computation to external tools, a paradigm that is becoming more and more frequent.
>
> We thank you again for the great comments that have helped strengthen our work. We hope that our additional experiments and discussion make you more confident about our contributions. We are of course happy to clarify any further points in more detail and we thank you for endorsing our paper.
>
> [1] The Llama 3 Herd of Models, July 23, 2024. https://ai.meta.com/research/publications/the-llama-3-herd-of-models/.
>
> [2] Train Short, Test Long: Attention with Linear Biases Enables Input Length Extrapolation, Press et al. Arxiv, 2021.
>
> [3] Attention is all you need, Vaswani et al. Neurips, 2017.
>
> [4] The Impact of Positional Encoding on Length Generalization in Transformers, Kazemnejad et al, Neurips, 2023.

---

> > ### Comment · Reviewer_SAa1 · 2024-08-09
> >
> > Thank you for your satisfactory comments to my questions. I will raise my score to 6.

---

> > > ### Author Response · Authors · 2024-08-11
> > > **Thank you!**
> > >
> > > We would like to thank you for acknowledging our responses and increasing your score. We remain available should you have any additional questions during the discussion period!

---

### Author Rebuttal · Authors · 2024-08-06

We are delighted to see that our paper has been well-received by all reviewers, with comments on the high quality of the writing and presentation.

We would like to summarize the improvements we have made to our paper. We have added a supplementary one-page PDF with the additional experiments in our response as well.

- Improved discussion and experiments regarding positional encodings, now also including Alibi [1], Absolute Sinusoidal Embeddings (APE) [2], and no positional embeddings (NoPE) [3] (Reviewers SAa1 and GKd9)
- Ablation on the structure of the prompt (Reviewer SAa1)
- Added discussion on windowed masked attention (Reviewer vhDf)
- Improved figure readability (Reviewer GKd9)

In this global comment we address in more detail topics that were touched upon by more than one reviewer.

**Effect of different positional encodings**

Reviewers SAa1 and GKd9 both asked how different positional encodings (PEs) play a role in representational collapse. The current experiments focus on RoPE, which is widely used in LLMs today, e.g. Gemma and Llama 3 [1].

The theory uses the fact that the effect of the positional encodings decreases with distance, which we believe applies to many of the popular encodings used today. To showcase this, we have designed a synthetic experiment testing the representational collapse using RoPE, Alibi [2], Absolute Sinusoidal Embeddings (APE) [3], and no positional embeddings (NoPE) [4]. The experiment serves as a very controlled setting to ablate positional encodings (absolute and relative) in isolation.

In our experiment, we sample the entries of the queries, keys, and values from a standard Gaussian independently and then apply the various encodings, taking into account normalisations due to e.g. layer norm. We sample sequences of length n and create sequences of length n+1 by taking the sequence of length n and repeating the last token. We then measure the L1 distance between the two latent vectors of the last token of the two sequences. The result can be seen in Figure 1 in the supplementary PDF, noting that the y-axis is measured in log-scale. We set the hidden dimension to 64, use a single layer, and focus on a single attention head. It is clear that the different positional encodings converge very similarly, supporting our theoretical claims on representational collapse.

**Practical implications of our work**

All reviewers have asked for us to comment further on the practical implications of our work. We believe that highlighting the brittleness of fundamental operations of a Transformer-based model is important to increase our understanding of the robustness of such models and to ultimately help us improve them. The prompts we chose allowed us to more carefully measure and support our theory, as general-purpose queries may have many more confounding variables.

We reiterate that the tasks we study (counting and copying) are essential for future agent-based AI systems. Especially in settings where AI off-loads most of the computation to other tools, it is important to be able to correctly copy the input into such tools.

We believe there are numerous real-world applications which this work could prove to be insightful for, particularly for tasks which see frequent repetition of tokens. One example is finance in which numbers could contain a large number of 0s (e.g. number of shares, company valuations, etc.), which was one of the motivations for us to test the introduction of “,” characters to delimit the repetitions. Other examples could be the processing of spreadsheets which could present many repeated values or LLMs operating over bit sequences.

Further, we are excited by the prospects of using our work to explain existing observed phenomena in LLMs, for instance, the “lost-in-the-middle” phenomenon. We hope that our work acts as a foundation to understand further issues related to representational collapse and over-squashing in LLMs.

We are very much looking forward to the rebuttal period and thank the reviewers for helping improve and strengthen our work with their valuable insights, questions, and comments.

[1] The Llama 3 Herd of Models, July 23, 2024. https://ai.meta.com/research/publications/the-llama-3-herd-of-models/.

[2] Train Short, Test Long: Attention with Linear Biases Enables Input Length Extrapolation, Press et al. Arxiv, 2021.

[3] Attention is all you need, Vaswani et al. Neurips, 2017.

[4] The Impact of Positional Encoding on Length Generalization in Transformers, Kazemnejad et al, Neurips, 2023.

---

### Decision · Program_Chairs · 2024-09-25

**Decision:**

Accept (poster)

**Comment:**

The paper indicates some major shortcomings suffered by Transformer architectures via both empirical studies and theoretical analysis. The work could be further improved if the authors could explain their discoveries with more practical examples (in contrast to the toy tasks used in the paper), and it would be even greater if the authors could provide some insights for how to remedy the shortcomings. Despite these weakness, all reviewers are positive to the work, and I also agree that the work sheds light on the future studies of Transformers. Therefore, I would like to recommend an acceptance.